# Mitochondrial H$_2$O$_2$ release does not directly cause damage to chromosomal DNA

Daan M. K. van Soest [1], Paulien E. Polderman[1], Wytze T. F. den Toom [1], Janneke P. Keijer [1], Markus J. van Roosmalen [2], Tim M. F. Leyten[1], Johannes Lehmann [1], Susan Zwakenberg[1], Sasha De Henau [1], Ruben van Boxtel [2,3], Boudewijn M. T. Burgering [1,3] & Tobias B. Dansen [1]✉

Reactive Oxygen Species (ROS) derived from mitochondrial respiration are frequently cited as a major source of chromosomal DNA mutations that contribute to cancer development and aging. However, experimental evidence showing that ROS released by mitochondria can directly damage nuclear DNA is largely lacking. In this study, we investigated the effects of H$_2$O$_2$ released by mitochondria or produced at the nucleosomes using a titratable chemogenetic approach. This enabled us to precisely investigate to what extent DNA damage occurs downstream of near- and supraphysiological amounts of localized H$_2$O$_2$. Nuclear H$_2$O$_2$ gives rise to DNA damage and mutations and a subsequent p53 dependent cell cycle arrest. Mitochondrial H$_2$O$_2$ release shows none of these effects, even at levels that are orders of magnitude higher than what mitochondria normally produce. We conclude that H$_2$O$_2$ released from mitochondria is unlikely to directly damage nuclear genomic DNA, limiting its contribution to oncogenic transformation and aging.

Genomic DNA damage and subsequent mutation and selection is not only a driver of evolution but also of oncogenesis, as it can result in the acquisition of the Hallmarks of Cancer[1]. A tight control of genome integrity is therefore required to suppress malignant cell growth. The cellular DNA damage response (DDR), for instance, triggers a temporary cell cycle arrest to repair damaged DNA. Persistent DNA damage induces apoptosis or a permanent withdrawal from the cell cycle (senescence), which is thought to be both a mechanism for tumor suppression as well as a major cause of organismal aging[2]. DNA damage results from chemical modification, including DNA-base oxidation, which subsequently leads to mutagenesis because of mispairing during replication or inadequate repair. The most abundant oxidative base modification is 8-oxo-deoxy-Guanine (8-oxo-dG)[3,4], which can mispair with Adenine. If left unrepaired, the oxidation of DNA on Guanine thus leads to C > A mutations upon replication, whereas incorporation of an oxidized dG nucleotide across from A leads to T > G mutations. Enzymes that repair or prevent oxidative DNA

lesions are found in all three domains of life (eukaryotes, bacteria & archaea), indicating that the prevention of genomic instability due to DNA oxidation is needed in many, if not all, organisms[5–7].

Mutagens that threaten DNA integrity do not only come from exogenous sources (like UV-light), but are also generated endogenously through cellular metabolism. Reactive Oxygen Species (ROS) are an example of potentially harmful metabolic products. ROS are generated at various intracellular sites in the form of superoxide (O$_2^{\cdot-}$) or hydrogen peroxide (H$_2$O$_2$)[8]. Importantly, both O$_2^{\cdot-}$ and H$_2$O$_2$ are not reactive with DNA[9]. However, H$_2$O$_2$ can generate highly reactive hydroxyl radicals ($^{\cdot}$OH), catalyzed by ferrous iron (Fe$^{2+}$) in the Fenton reaction, and the latter has been shown to be able to cause various oxidative DNA lesions in vitro[3,4]. The respiratory chain that drives mitochondrial ATP production is one of the major sites of metabolic ROS generation within the cell[10]. Respiratory chain derived ROS are generated in the form of O$_2^{\cdot-}$, which are instable and very short-lived. O$_2^{\cdot-}$ is rapidly converted to the more stable H$_2$O$_2$ by superoxide

[1]Center for Molecular Medicine, University Medical Center Utrecht, Universiteitsweg 100, Utrecht 3584 CG, The Netherlands. [2]Princess Máxima Center for Pediatric Oncology, Heidelberglaan 25, Utrecht 3584 CS, The Netherlands. [3]Oncode Institute, Jaarbeursplein 6, Utrecht 3521 AL, The Netherlands. ✉e-mail: t.b.dansen@umcutrecht.nl

dismutases located in the cytosol and the mitochondrial inter-membrane space (SOD1) and the mitochondrial matrix (SOD2). The combined findings that exposure of cells to extracellular ROS triggers the DDR, that $H_2O_2$ (via ·OH) can oxidize DNA in vitro, and that mitochondria are one of the major intracellular suppliers of $H_2O_2$ are often extrapolated to the notion that ROS production and release by mitochondria leads to oxidative DNA damage and mutation[11–13]. This idea is also firmly rooted in the general public and is probably one of the foundations under the popular use of dietary antioxidant supplements. But evidence for direct oxidation of nuclear, genomic DNA by mitochondria-derived ROS in live cells is largely lacking. As the diffusion range of ·OH is very limited due to its extreme reactivity, it would need to be formed close to the DNA to cause oxidative DNA-base modifications. This strongly suggests that DNA-base oxidation by mitochondrial ROS could only occur in case mitochondria-derived $H_2O_2$ would diffuse into the nucleus and form ·OH in proximity to the DNA. However, recent studies show that steep $H_2O_2$ gradients exist around the mitochondria due to efficient scavenging by peroxidases in human, yeast as well as plant cells[14–17], making it unlikely that $H_2O_2$ can act over distances large enough to reach the nucleus when released from mitochondria. Indeed, Cockayne syndrome-B (CS-B) mutated fibroblasts display elevated levels of mitochondrial ROS, but this does not translate into increased nuclear DNA damage[18]. On the other hand, when cells are cultured under physiological rather than atmospheric oxygen tension, far less mutations downstream of DNA-base oxidation are being acquired[19]. Exogenously added $H_2O_2$ can also trigger the DNA damage response[20], but it is not clear how this compares to physiological continuous production of $O_2^{·-}$ and subsequently $H_2O_2$ by mitochondria in terms of achieved intracellular steady state and peak concentrations. Taken together, it remains unclear to what extent respiration-derived ROS contribute directly to DNA damage.

Recently, ectopic expression of the enzyme D-amino acid oxidase (DAAO) from the yeast *Rhodotorula gracilis* has been used to achieve titratable and sustained intracellular $H_2O_2$ production in live human cells as well as in in vivo rodent models[14,21,22]. $H_2O_2$ production is induced only upon administration of D-amino acids like D-Alanine (D-Ala), which are (largely, if not completely) absent from cultured cells. DAAO can be targeted to the organelle of choice by fusion to localization tags. $H_2O_2$ production by DAAO is often assessed using genetically encoded ratiometric fluorescent redox sensors like HyPer7[14,15]. Although these sensors are highly sensitive to $H_2O_2$, they report on the combined rate of oxidation and reduction, preventing direct quantification of absolute levels of $H_2O_2$ produced. To overcome this, we have recently developed a method to quantify DAAO dependent $H_2O_2$ production based on oxygen consumption rates, which allows for the careful titration of D-Ala levels that yield near and supraphysiological levels of $H_2O_2$[23].

We set out to systematically analyze the localization-dependent cellular response to $H_2O_2$, making use of the chemogenetic production of $H_2O_2$ by ectopic expression of DAAO. To mimic mitochondrial $H_2O_2$ release, we fused DAAO to the cytosol-facing end of the targeting sequence of the outer mitochondrial membrane protein TOMM20 and stably expressed it in RPE1-hTERT cells. RPE1 cells expressing nucleosome-targeted DAAO serve as a positive control for what happens when $H_2O_2$ reaches the nuclear DNA. Careful titration of a substrate for DAAO (D-Ala) enables the continuous production of near and supra-physiological $H_2O_2$ levels. Using this approach, we show that $H_2O_2$ released from mitochondria does not induce direct damage to the nuclear DNA when produced at levels likely achievable by respiration. In contrast, $H_2O_2$ generation in close proximity to the nuclear DNA causes DNA mutations, DNA strand breaks and subsequent activation of the DNA damage response (DDR), resulting in cell cycle arrest with many characteristics of senescence, similar to what has been described for cells exposed to high levels of exogenous ROS. Based on these observations we conclude that mitochondrial respiration-derived ROS is probably not a major factor in the induction of chromosomal DNA mutations, in both non-transformed human RPE1-hTERT and MCF7 breast cancer cells.

## Results

### Stable expression of D-amino acid oxidase (DAAO) in RPE1-hTERT enables localized intracellular $H_2O_2$ production

The various effects of ROS on cells have mostly been studied by applying a bolus (typically 10–200 μM) of $H_2O_2$ to the tissue culture medium. Although such treatments certainly induce oxidative stress, it does not recapitulate the continuous ROS production (i.e. $O_2^{·-}$ and subsequently $H_2O_2$) by mitochondria, which occurs at far lower levels. We therefore made use of chemogenetic production of $H_2O_2$ using localized DAAO expression. For this study we generated monoclonal RPE1-hTERT cell lines stably expressing mScarlet-I-DAAO localized to different sites (Fig. 1a). H2B-mScarlet-I-DAAO localizes to nucleosomes to study the effects of $H_2O_2$ when it is produced in close proximity to DNA. TOM20-mScarlet-I-DAAO localizes to the cytosolic side of the outer mitochondrial membrane and hence mimics the release of $H_2O_2$ by mitochondria. RPE1-hTERT cells were chosen as this human cell line is untransformed and the signal transduction pathways relevant for cell cycle regulation and the DNA damage response are fully functional. We will refer to these cell lines from now on as RPE1-hTERT-DAAO^H2B and RPE1-hTERT-DAAO^TOM20. Imaging of mScarlet-I confirms the expected localization of DAAO by colocalization with DNA (Hoechst 33342 staining) for RPE1-hTERT-DAAO^H2B cells and mitochondria (MitoTracker Green staining) for RPE1-hTERT-DAAO^TOM20 cells respectively (Fig. 1b).

Total DAAO activity in these cell lines is likely determined by a combination of the level of DAAO expression, folding, accessibility, cofactor (FAD) availability as well as substrate (D-amino acid) levels, uptake, and diffusion[24]. Because DAAO uses equimolar amounts of molecular oxygen to produce $H_2O_2$, we assessed DAAO activity by measuring the oxygen consumption rate (OCR) in response to titration with [D-Ala] as we described before[20]. Mitochondrial respiration of RPE1-hTERT-DAAO^H2B and RPE1-hTERT-DAAO^TOM20 cells was inhibited by oligomycin prior to D-Ala addition to be able to compare DAAO-derived $H_2O_2$ to mitochondrial OCR. In this way monoclonal RPE1-hTERT-DAAO^TOM20 and RPE1-hTERT-DAAO^H2B lines with similar DAAO activities were identified and selected (Fig. 1c, d). Thus, observed differences between these lines upon DAAO activation are the result of differential localization of $H_2O_2$ production and not of total oxidative burden. Both cell lines consume an amount of oxygen equivalent to roughly 10% of basal mitochondrial respiration at 5 mM D-alanine, which comes down to about 0.4 fmol $H_2O_2$ cell$^{-1}$ min$^{-1}$ [23]. It is often stated that about 2% of oxygen escapes the mitochondrial ETC as $O_2^{·-}$, although this number is based on maximum $O_2^{·-}$ production by complex III under saturating oxygen and substrate conditions using isolated mitochondria, whereas $O_2^{·-}$ production by the ETC in live cells has been suggested to be ten times lower (reviewed in[25]). Assuming that somewhere between 0.2% and 2% of the mitochondrial consumed oxygen becomes $O_2^{·-}$ and subsequently $H_2O_2$ and taking into account that 2 molecules of $O_2^{·-}$ form 1 molecule of $H_2O_2$ catalyzed by SODs, an increase in OCR upon D-Ala addition of 10% of mitochondrial OCR represents 10–100x the normal mitochondrial $H_2O_2$ production. Hence, at 5 mM of D-Ala, mitochondrial $H_2O_2$ release as modeled by the RPE1-hTERT-DAAO^TOM20 cell line is already far above what can be expected to occur under (patho)physiological conditions. In addition, while in our model system the $H_2O_2$ is directly produced on the cytosolic side of mitochondria, endogenous mitochondrial $H_2O_2$ production takes place inside mitochondria, and only a fraction of it will be released from the matrix and the IMS into the cytosol due to scavenging by for instance PRDX3[14–17]. Hence, at 5 mM of D-Ala, mitochondrial $H_2O_2$ release as modeled by the RPE1-hTERT-DAAO^TOM20 cell

line is already far above what can be expected to occur under (patho) physiological conditions.

To investigate how DAAO activation affects the thioredoxin-dependent redox status, we performed non-reducing western blots for the $H_2O_2$ scavenger peroxiredoxin 2 (PRDX2) (Supplementary Fig. 1). PRDX2 is found in the cytosol as well as the nucleus of cells. Upon oxidation of its $H_2O_2$-sensitive cysteines PRDX2 forms homodimers, which can be detected by an upward band shift on a non-reducing gel. Indeed, increasing amounts of D-Alanine resulted in increased dimerization of PRDX2. The [D-Ala] dependent induction of dimerization was more or less similar comparing RPE1-hTERT-DAAO[H2B] and RPE1-hTERT-DAAO[TOM20] cells, which is in line with OCR measurements (Fig. 1c, d). At high levels of $H_2O_2$, peroxiredoxins can become inactivated through overoxidation (PRDXSO[2/3]). In contrast to exogenous $H_2O_2$ treatment, no overoxidation of PRDX was detected upon DAAO activation at viable levels. This indicates that continues intracellular $H_2O_2$ production evokes a very different cellular redox response as compared to a bolus of exogenous $H_2O_2$.

## Diffusion of locally produced $H_2O_2$ is limited

For respiration-derived ROS to directly inflict damage to chromosomal DNA, it would need to diffuse out of mitochondria, across the cytosol and over the nuclear membrane. $O_2^{\cdot-}$ has a very limited diffusion range due to its low membrane permeability[26] and rapid conversion to $H_2O_2$ by Superoxide dismutases[27]. $H_2O_2$ has been suggested to be stable enough to travel some distance in biological systems, although the

exact range is debated. We assessed the diffusion range of $H_2O_2$ produced by DAAO at the outer mitochondrial membrane and at the nucleosome using live fluorescence microscopy of the ultrasensitive, ratiometric $H_2O_2$ sensor HyPer7 stably expressed in the nucleus (NLS-HyPer7) and the cytosol (NES-HyPer7). Oxidation of cytosolic HyPer7 was readily detected at low [D-Ala] in RPE1-hTERT-DAAO[TOM20] cells, and nuclear HyPer7 rapidly responded to low [D-Ala] in RPE[H2B-DAAO] cells, showing that HyPer7 is oxidized when in close proximity to the site of $H_2O_2$ production (Fig. 2a, b). Increased HyPer7 oxidation in response to D-Ala could be observed for up to 48 hrs after addition, although after ~10 hrs oxidation starts to drop (Supplementary Fig. 2A). Up until 10 mM of D-Ala, $H_2O_2$ produced by mitochondrial membrane-localized DAAO did not result in oxidation of nuclear HyPer7 over the time of measurement (16 hrs) (Fig. 2c). The (lack of) oxidation of NLS-HyPer7 in response to D-Ala treatment of RPE1-hTERT-DAAO[TOM20] was highly homogenous and there was no subset of cells that did show an increase (Supplementary Fig. 3). Only at very high DAAO substrate levels ([D-Ala] ≥ 20 mM, Supplementary Fig. 2B) a minor increase in NLS-HyPer7 was detected in RPE1-hTERT-DAAO[TOM20] cells. However, cells were not able to survive this amount of $H_2O_2$ production at the outer mitochondrial membrane for prolonged duration (Fig. 2e, Supplementary Fig. 4A). A similar trend was observed for the combination of NES-HyPer7 in RPE1-hTERT-DAAO[H2B] cells: No HyPer7 oxidation is detected outside the nucleus except when high amounts of DAAO substrate are supplied ([D-Ala] ≥ 10 mM, Fig. 2d, Supplementary Fig. 2B, Supplementary Fig. 3B), although there seemed to be some leakage of nuclear

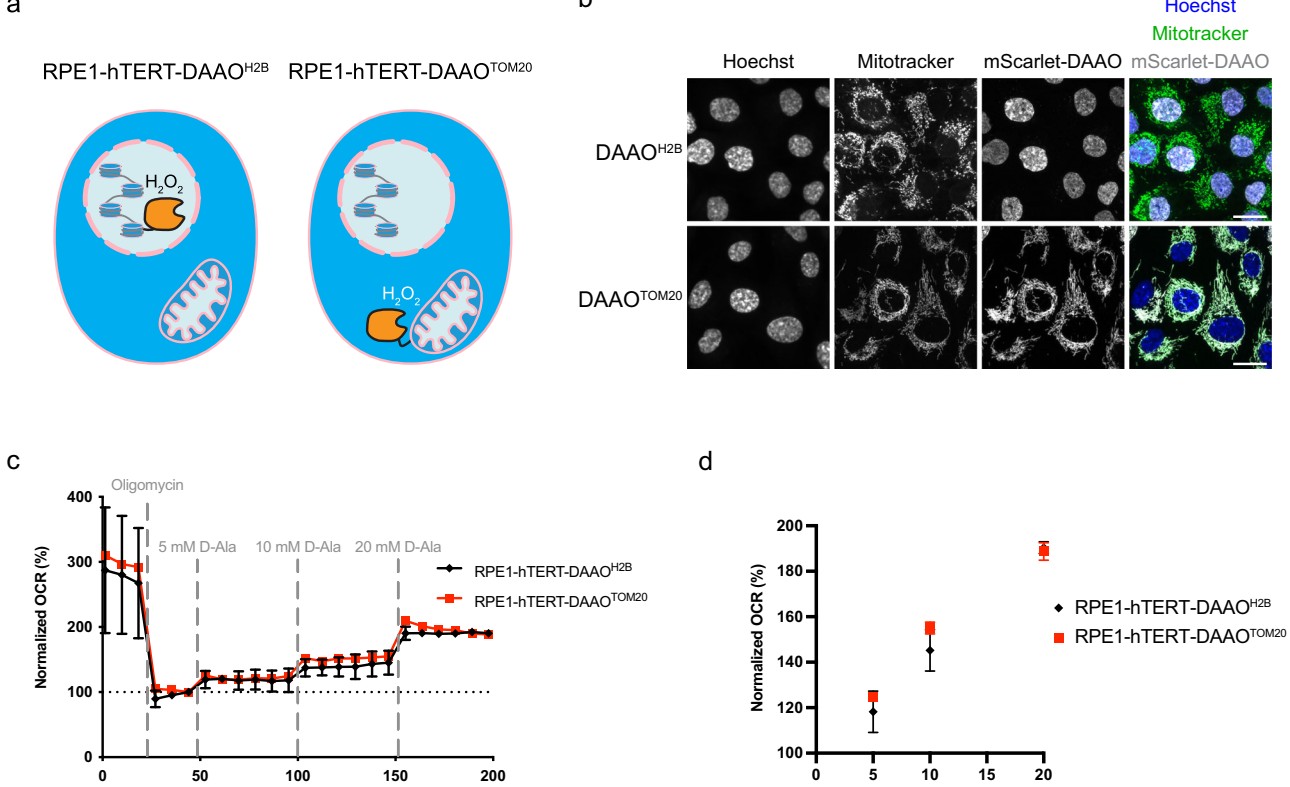

**Fig. 1 | Localized expression of DAAO enables intracellular $H_2O_2$ production.**
**a** Schematic representation of RPE1[hTERT] cells expressing H2B-mScarlet-I-DAAO or TOM20-mScarlet-I-DAAO (further described as DAAO[H2B] and DAAO[TOM20] respectively) to induce localized intracellular $H_2O_2$ production. **b** Immunofluorescence confocal images of cell lines depicted in (**a**) showing colocalization of DAAO[H2B] and DNA (Hoechst), and colocalization of DAAO[TOM20] with mitochondria (MitoTracker Green). Scale bar = 20 µm. **c** Oxygen consumption rate (OCR) upon D-Ala administration as a readout for enzymatic activity of ectopically expressed DAAO. The

increase in OCR shows that both lines produce roughly similar amounts of $H_2O_2$ upon addition of D-Ala. Data is normalized to OCR upon Oligomycin addition, at the last timepoint before D-Ala injection. The experiment was repeated at least 3 times with reproducible results, 1 biological repeat is depicted. Data are presented as mean values of 5 separate wells -/+ SD. Source data are provided in the Source Data file. **d** Normalized OCR values from (**c**) plotted versus [D-ala]. The 6th timepoint after each D-ala injection was used for this graph. Data are presented as mean values -/+ SD. Source data are provided in the Source Data file.

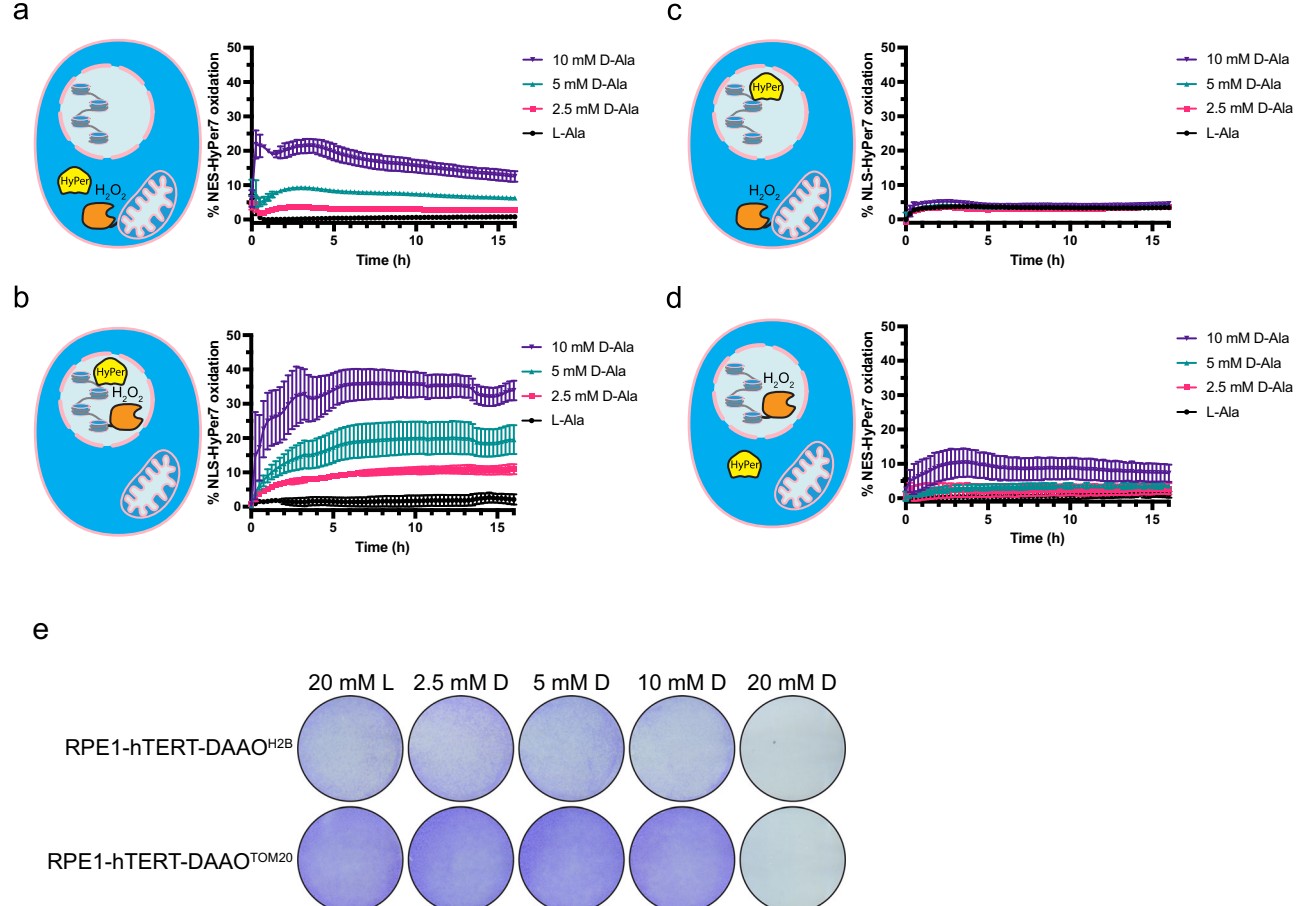

**Fig. 2 | H₂O₂ released from mitochondria is not detected in the nucleus at levels that are compatible with cell survival. a–d** H₂O₂ measurements using the ratiometric HyPer7 sensor to determine diffusion of DAAO-produced H₂O₂. Treating cells with 400 μM H₂O₂ was used to determine 100% HyPer7 oxidation. Lowest oxidation measured was set to 0%. NES-HyPer7, localized in the cytosol, becomes already oxidized at the lowest concentration (2.5 mM) of D-Ala used in RPE1-hTERT-DAAO$^{TOM20}$ cells (**a**) In contrast, nuclear NLS-HyPer7 oxidation is not detected at 0−10 mM D-Ala. **b** NLS-HyPer7 oxidation is already detected at 2.5 mM D-Ala in

RPE1-hTERT-DAAO$^{H2B}$ cells (**c**) while NES-HyPer7 oxidation is only detected starting from 10 mM D-Ala. **d** Data are presented as mean values -/+ SD, *n* = 3 biological replicates. Source data are provided in the Source Data file. **e** Representative image of crystal violet staining of DAAO lines after 24 h L/D-Ala treatment. Both RPE1-hTERT-DAAO$^{H2B}$ and RPE1-hTERT-DAAO$^{TOM20}$ cells do not survive continuous D-Ala treatments above 10 mM D-Ala. Experiment was repeated at least 3 times, a representative image is shown.

H₂O₂ into the cytosol already at 10 mM of D-Ala. Also, in the RPE1-hTERT-DAAO$^{H2B}$ cell line 20 mM of D-Ala lead to levels of H₂O₂ production that are not compatible with cell survival (Fig. 2e, Supplementary Fig. 4A). D-Ala treatment did not affect viability in RPE1-hTERT cells not expressing DAAO (Supplementary Fig. 4B). These observations are in line with earlier studies that suggested that in cells H₂O₂ is mostly confined to the site of production. Furthermore, when H₂O₂ is produced at levels high enough to diffuse from the cytosol into the nucleus this is incompatible with cell survival. Note that based on our OCR experiments (Fig. 1c, d) it can be estimated that the amount of H₂O₂ production needed to achieve diffusion from the outer mitochondrial membrane into the nucleus corresponds to >25% of oxygen used in basal mitochondrial respiration, which is difficult to imagine occurring under (patho)physiological conditions. We then tested whether mitochondria-derived H₂O₂ could reach the nucleus when the PRDX-TRX dependent H₂O₂ scavenging system was impaired by pretreatment with the Thioredoxin reductase inhibitor Auranofin. Auranofin pretreatment indeed led to higher cytosolic HyPer7 oxidation, but only at 10 mM of D-Ala a slight increase in nuclear HyPer7 oxidation could now be observed in the RPE1-hTERT-DAAO$^{TOM20}$(Supplementary Fig. 5), which corresponds to mitochondrial H₂O₂ release far above what can be expected physiologically. Higher concentrations of D-Ala rapidly killed the cells when combined with Auranofin pretreatment

and could therefore not be measured. Based on these observations and considerations we propose that it is highly unlikely that under physiological circumstances respiration-derived ROS can diffuse into the nucleus and directly induce oxidative damage to chromosomal DNA that could lead to mutations.

## Mitochondrial H₂O₂ release does not induce nuclear DNA damage

Although the HyPer7 sensor is very sensitive, it could be that undetectable levels of H₂O₂ released from mitochondria make it into the nucleus and damage the DNA. Although H₂O₂ derived from mitochondria may not travel to the nucleus and cause DNA damage by direct oxidation, DNA damage downstream of H₂O₂ could in principle occur through mechanisms that would not require diffusion of H₂O₂ into the nucleus. Firstly, cytosolic H₂O₂ could lead to oxidation of the nucleotide pool and subsequent incorporation of these oxidized nucleotides in the DNA during replication. Secondly, cytosolic H₂O₂ may hamper the fidelity or activity of proteins involved in DNA repair and replication by direct or indirect oxidation. The DNA repair protein RAD51 for instance has been shown to be inhibited upon Cysteine oxidation[28]. Treatment of cells with a high dose of exogenous H₂O₂ has been shown to activate both the ataxia telangiectasia and RAD3-related protein (ATR) and the ataxia telangiectasia mutated protein (ATM)

dependent DNA damage response (DDR), indicative of single and double strand DNA breaks respectively, and this is also the case in the used RPE1-hTERT cell line[20,28]. Repair of 8-oxo-dG in nuclear genomic DNA proceeds through a transient single strand break and may even lead to double strand breaks[29]. Therefore, we monitored whether the DDR was activated in RPE1-hTERT-DAAO^TOM20 and RPE1-hTERT-DAAO^H2B lines upon D-Ala treatment. Indeed, induction of nuclear $H_2O_2$ production by DAAO^H2B resulted in a clear activation of ATR and ATM, indicated by phosphorylation of their downstream targets checkpoint kinases CHK1 and CHK2 respectively, which was observed 2 h following addition of D-Ala (Supplementary Fig. 6A) and sustained for at least 48 hrs (Fig. 3a). This coincided with the stabilization of p53 and subsequent transcription of the cell cycle inhibitor p21. Knockout of p53 resulted in even higher levels of pCHK1, pCHK2 and γH2AX, likely due to a sustained DDR because of impaired DNA repair. However, without p53 this was not translated into transcription of p21. In contrast to $H_2O_2$ produced at the nucleosome, mimicking the release of mitochondrial $H_2O_2$ by DAAO^TOM20 did not result in activation of the DDR, even when 10 mM of D-Ala was supplied for 48 hrs (Fig. 3b, Supplementary Fig. 6B). Even in the absence of p53, no increase in CHK1 or CHK2 phosphorylation or γH2AX levels was observed (Fig. 3b).

DAAO^TOM20 mimics mitochondrial $H_2O_2$ release, but mitochondrial ROS is normally produced at both sides of the inner mitochondrial membrane by Electron Transport Chain[20]. We therefore tested whether the DNA damage response was perhaps activated in response to production of $H_2O_2$ in the mitochondrial matrix (RPE1-hTERT-DAAO^MLS) or the intermembrane space (RPE1-hTERT-DAAO^IMS), but this was not the case (Supplementary Fig. 7).

When cells go through mitosis the nuclear envelope, which could be a barrier for $H_2O_2$ diffusion into the nucleus[23], is broken down. Mitotic cells also round up, which shortens the average distance of mitochondria to the chromosomes[23]. We therefore wondered whether mitochondria-derived $H_2O_2$ could induce DNA damage in mitotic cells. Cells were trapped in metaphase by nocodazole treatment (Supplementary Fig. 8A), followed by D-Ala treatment to start $H_2O_2$ production. Under these conditions the DDR was again only activated in RPE1-hTERT-DAAO^H2B but not in RPE1-hTERT-DAAO^TOM20 cells (Supplementary Fig. 8B). Hence, high levels of $H_2O_2$ production at the outer mitochondrial membrane does not induce DNA damage as measured by activation of the DDR, even in the absence of the nuclear envelope and when mitochondria are closer to the DNA.

Activation of the DDR is an indirect measurement of DNA damage. To test whether DNA single and double strand breaks occur downstream of localized $H_2O_2$ production in a more direct way, we performed alkaline comet assays. RPE1-hTERT-DAAO^H2B lines showed a dose-dependent increase in DNA breaks upon D-Ala treatment, indicating that $H_2O_2$ indeed can induce DNA strand breaks (Fig. 3c, d). Note that DNA breaks can be detected at levels of DAAO^H2B-induced $H_2O_2$ that are not overtly toxic (e.g., at [D-Ala] ≤ 10 mM), suggesting that this amount of damage is still repairable. In contrast, no increase in DNA strand breaks was detected when RPE1-hTERT-DAAO^TOM20 were treated with [D-Ala] compatible with cell survival (Fig. 2e, Supplementary Fig. 4A). At 40 mM D-Ala DNA strand breaks can be detected in RPE1-hTERT-DAAO^TOM20 cells (Fig. 3c, d), but again it is unlikely that these levels of $H_2O_2$ can be achieved physiologically, as this corresponds to an amount of $H_2O_2$ production ~two orders of magnitude higher than what mitochondria normally produce (Fig. 1c). Besides, this level of $H_2O_2$ production at the mitochondria invariably kills the cells, which precludes mutation accumulation down the lineage (Fig. 2e, Supplementary Fig. 4A).

To corroborate these observations in another cell line, we devised monoclonal MCF7 human breast cancer cell lines expressing DAAO^H2B and DAAO^TOM20 with similar DAAO activities as measured by OCR (Supplementary Fig. 9A, Supplementary Fig. 9B). MCF7-DAAO^H2B and

MCF7-DAAO^TOM20 cells were a bit more sensitive to $H_2O_2$ production and viability dropped dramatically at 48 hrs of [D-Ala] >5 mM (Supplementary Fig. 9C). DNA damage was induced as measured by DDR activation (Supplementary Fig. 9D) and the comet assay (Supplementary Fig. 9F) using 5 mM D-Ala in MCF7-DAAO^H2B but not in MCF7-DAAO^TOM20 cells (Supplementary Fig. 9E, Supplementary Fig. 9F), indicating that also in this breast cancer cell line mitochondrial $H_2O_2$ release does not directly induce nuclear DNA damage.

Oxidative DNA damage also occurs at the single base level, and it could hypothetically be the case that low levels of mitochondria-derived $H_2O_2$ that are not detected by NLS-HyPer7 could induce DNA damage in the absence of extensive DNA strand breaks and is therefore not detected by Western blot for activation of the ATM and ATR-dependent DDR or by the Comet assay (Fig. 2a–d). To be able to pick up DNA damage and subsequent mutation with single-base resolution we performed Whole Genome Sequencing (WGS) on RPE1-hTERT-DAAO^H2B and RPE1-hTERT-DAAO^TOM20 following D-Ala treatment and performed mutational analysis. Because mutations occur randomly in the genome and hence average out in bulk sequencing, monoclonals must be derived after D-Ala treatment in order to call variants. To potentially induce many mutations, but without preventing monoclonal outgrowth of the cells by loss of viability, we performed the WGS experiment in a p53 KO background (see also further in this study) and subjected the cells to 4 rounds of 2 hrs 20 mM D-Ala followed by 22 hrs recovery (Fig. 3e). Again, this level of D-Ala produces far more $H_2O_2$ than what likely can be achieved by mitochondria, but with 2 hrs treatment followed by recovery it does not overtly affect viability (Supplementary Fig. 10, Supplementary Fig. 11). In line with the induction of the DDR and results from the comet assay, many point mutations were induced in the RPE1-hTERT-DAAO^H2B p53 KO cells (Fig. 3F). No significant increase in the number of point mutations was observed in RPE1-hTERT-DAAO^TOM20 p53 KO cells upon repeated D-Ala treatment (Fig. 3f).

Collectively, these experiments show that $H_2O_2$ produced in close proximity to the DNA can indeed trigger widespread DNA damage and mutation, DNA breaks and a subsequent DNA damage response. $H_2O_2$ released by mitochondria is not able to induce these phenotypes at levels compatible with cellular survival.

## A p53-dependent cell cycle arrest and senescence is induced by $H_2O_2$ production in proximity to DNA but not by mitochondrial $H_2O_2$ release

Exogenous addition of $H_2O_2$ induces a cell cycle arrest in RPE1-hTERT cells[30]. To investigate whether localized $H_2O_2$ production can also trigger a cell cycle arrest, and whether that depends on its site of production, we performed a BrdU incorporation assay. RPE1-hTERT-DAAO^TOM20 and RPE1-hTERT-DAAO^H2B cells were treated for 48 h with L- or D-Ala, followed by incubation in medium containing BrdU (and no L- or D-Ala) for 24 h to label cycling cells. BrdU positive and negative cells were assessed by flow cytometry. In line with our observation that $H_2O_2$ production at the nucleosome triggers p53 stabilization and activity, $H_2O_2$ production in RPE1-hTERT-DAAO^H2B cells resulted in a clear loss of proliferative capacity upon treatment with D-Ala. $H_2O_2$ release from mitochondria on the other hand failed to induce a cell cycle arrest at any concentration that did not result in cell death (Fig. 4a). In line with the observed stabilization and activation of p53 upon $H_2O_2$ production at the nucleosome (Fig. 3a), the observed arrest in RPE1-hTERT-DAAO^H2B cells was indeed p53 dependent (Fig. 4b). We then assessed the ploidy of the arrested cells by cell cycle profiling by flow cytometry. RPE1-hTERT-DAAO^H2B cells with 4 N DNA accumulated upon induction of $H_2O_2$ production, whereas the cell cycle profile of RPE1-hTERT-DAAO^TOM20 cells remained unchanged (Fig. 4c, d), and this was similar in the MCF7 breast cancer cell line (Supplementary Fig. 9G). The 4 N arrest was somewhat less pronounced in the MCF7-DAAO^H2B compared to the RPE1-hTERT-DAAO^H2B, which could be due to the fact

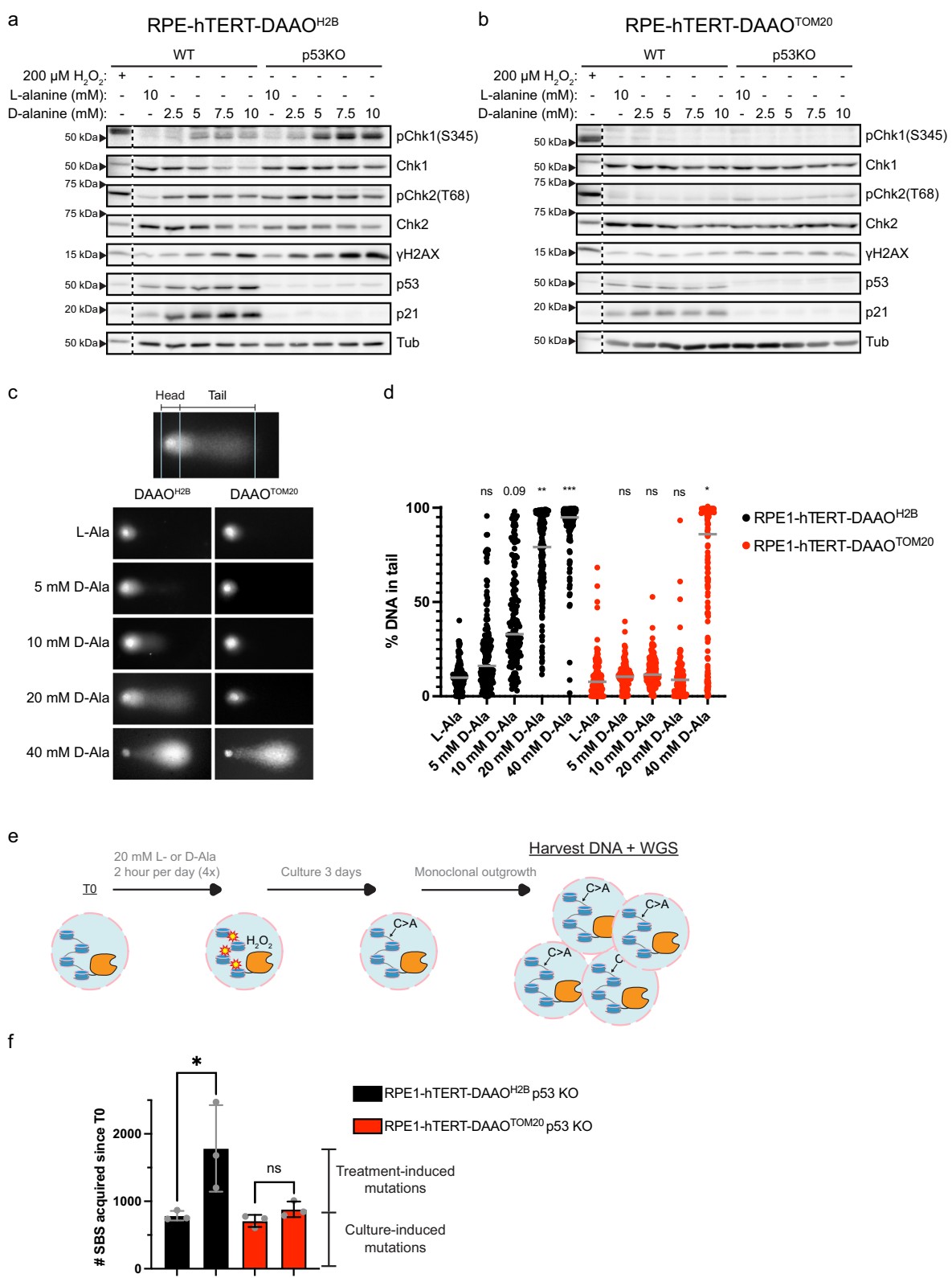

that MCF7 cells are tumor cells, which in general have weakened cell cycle checkpoints. The induction of the arrest with a ploidy of 4 N in RPE1-hTERT-DAAO[H2B] cells was largely p53-dependent (Supplementary Fig. 12). To test whether part of the cells also arrest with 2 N DNA, we performed a similar experiment but trapped cycling cells in mitosis (4 N DNA) using the spindle poison nocodazole during the last 16 h of D-Ala treatment, in order to more easily discriminate cells arrested

with 2 N DNA from the majority of cycling cells that reside transiently in $G_1$ in a standard cell cycle profile. In line with the results from the BrdU incorporation experiment (Fig. 4a), there was no sign of a cell cycle arrest with 2 N DNA in RPE1-hTERT-DAAO[TOM20] cells (Fig. 4e). There was a slight increase in cells with 2 N DNA in RPE1-hTERT-DAAO[H2B] cells treated with 2.5 mM D-Ala (Fig. 4e). We conclude that the majority of RPE1-hTERT-DAAO[H2B] cells arrest with 4 N DNA in a

**Fig. 3 | DNA damage, mutations and activation of the DNA damage response are induced by nuclear $H_2O_2$ but not by mitochondrial $H_2O_2$ release. a, b** Western Blot for components of the DNA damage response in WT or p53KO RPE1-hTERT-DAAO[H2B] and RPE1-hTERT-DAAO[TOM20] cells treated for 48 h with L- or D-alanine. Treatment for 15 min with 200 μM of exogenous $H_2O_2$ serves as a positive control. Nuclear $H_2O_2$ production results in activation of the DNA damage response (pCHK1, pCHK2, γH2AX) and subsequent p53 stabilization and p21 expression. In contrast, $H_2O_2$ produced at mitochondria by DAAO[TOM20] does not activate the DNA damage response. Similar results have been obtained at least 3 times, a representative experiment is shown. Source data are provided in the Source Data file. **c, d** Representative images and quantification of alkaline comet assay using RPE1-hTERT-DAAO[H2B] and RPE1-hTERT-DAAO[TOM20] cells treated for 2 h with L- or D-Ala. $H_2O_2$ production at mitochondria does not result in DNA strand breaks at levels that are compatible with survival (≤10 mM D-Ala). However, $H_2O_2$ production close

to the DNA does give rise to DNA strand breaks. Quantifications display all data points of 3 biological repeats along with the median. For each repeat ~50 comets/condition were analyzed. One-way Anova analysis + Dunnett's multiple comparisons test is performed on the mean of the 3 biological repeats per cell line, conditions are compared to L-Ala controls. (*$p ≤ 0.05$, **$p ≤ 0.01$, ***$p ≤ 0.001$, 20 mM D-Ala = 0.0013, 40 mM D-Ala = 0.002, DAAO[TOM20] 40 mM D-Ala = 0.0133). Source data are provided in the Source Data file. **e** Schematic of workflow for obtaining DNA samples for whole genome sequencing (WGS) and mutational analysis. **f** Quantification of the amount of single base substitutions (SBS) acquired in RPE1-hTERT-DAAO[H2B];p53KO and RPE1-hTERT-DAAO[TOM20];p53KO cells treated with L-Ala or D-Ala. $n = 3$ biological independent clones per condtion; Data are presented as mean values -/+ SD. A one-sided unpaired $t$-test was performed for each line to compare L-Ala to D-Ala treatment (*$p ≤ 0.05$, DAAO[H2B] L-Ala vs. D-Ala = 0.0277). Source data are provided in the Source Data file.

p53-dependent manner in response to viable levels of DAAO activation, whereas the RPE1-hTERT-DAAO[TOM20] cells keep cycling.

An arrest with 4 N DNA does not necessarily mean that the cells are arrested in $G_2$ or M phase of the cell cycle: previous studies have shown that $H_2O_2$ treatment can induce mitotic bypass after which cells permanently exit the cell cycle with 4 N DNA[31–33]. We used the FUCCI (Fluorescent Ubiquitin-based Cell Cycle Indicator) system[34] combined with staining for DNA-content and found that the RPE1-hTERT-DAAO[H2B] cells with 4 N DNA were indeed arrested in $G_{0/1}$ (Fig. 4f) and video timelapse microscopy showed that many cells indeed progress from $G_2$ directly to $G_{0/1}$ without passing through mitosis (Supplementary Movie 1, Supplementary Movie 2, Supplementary Fig. 13). The observed arrest with 4 N DNA in RPE1-hTERT-DAAO[H2B] cells was maintained despite D-Ala being washed away before BrdU incorporation and we tested whether these cells indeed are permanently arrested and acquired markers for senescence. Cells were kept in culture for an additional 5 days after treatment and washing out D-Ala to allow expression of senescence markers. RPE1-hTERT-DAAO[H2B] cells indeed gained senescence associated-β-galactosidase activity upon treatment with D-Ala (Fig. 5a, b). Accordingly, immunofluorescence microscopy indicated that RPE1-hTERT-DAAO[H2B] cells accumulated p21, lost Lamin B1 and displayed senescence-associated heterochromatin foci (SAHFs), all of which are makers of senescent cells[34,35](Fig. 5c, d, Supplementary Fig. 14). Based on (1) absence of BrdU positivity, (2) accumulation in $G_{0/1}$ with 4 N DNA following mitotic bypass, (3) sustained p21 induction, (4) loss of lamin B1, and (5) positivity for senescence associated beta-galactosidase, we conclude that $H_2O_2$ produced in the proximity of the DNA in RPE1-hTERT-DAAO[H2B] cells induces a cell cycle arrest that is reminiscent of senescence, similar to what was previously shown to be induced by bolus exogenous $H_2O_2$[34]. None of these phenotypes are induced in RPE1-hTERT-DAAO[TOM20] upon D-Ala treatment.

### Supraphysiological mitochochondrial $H_2O_2$ release triggers cell death

$H_2O_2$ production at the mitochondrial outer membrane in RPE1-hTERT-DAAO[TOM20] cells does not trigger a cell cycle arrest (Fig. 4a), but at high concentrations of D-Ala these cells did no longer grow out, even in a p53 KO background (Fig. 2e, Supplementary Fig. 10, Supplementary Fig. 11). We assessed whether mitochondrial function is compromised in response to $H_2O_2$ production in RPE1-hTERT-DAAO[TOM20] cells (as well as RPE1-hTERT-DAAO[IMS] and RPE1-hTERT-DAAO[MLS] cells, Supplementary Fig. 15A), but we found no evidence of changes in mitochondrial morphology or membrane potential up until 10 mM of D-Ala treatment in RPE1-hTERT-DAAO[TOM20] cells (Supplementary Fig. 15B, Supplementary Fig. 15C). In RPE1-hTERT-DAAO[IMS] and RPE1-hTERT-DAAO[MLS] cells however, mitochondria looked more fragmented upon D-Ala treatment and TMRM staining showed that the mitochondrial membrane potential dropped starting from 2.5 mM D-Ala. Likewise, the Seahorse mito-stress test performed 24 hrs after

D-Ala treatment showed that basal and maximal mitochondrial respiration in RPE1-hTERT-DAAO[IMS] and RPE1-hTERT-DAAO[MLS] cells dropped dramatically, whereas mitochondrial respiration was unaffected in RPE1-hTERT-DAAO[TOM20] cells (Supplementary Fig. 15D). These observations indicate that $H_2O_2$ produced by DAAO at the cytosolic side of the mitochondrial membrane does not lead to loss of mitochondrial function, and is therefore likely not the cause for the loss of viability upon higher levels of D-Ala.

We wondered whether these cells could succumb by ferroptosis, which can be triggered when $H_2O_2$ forms hydroxyl radicals in the presence of iron. Indeed, pre-treatment with the ferroptosis inhibitors ferrostatin or liproxstatin could partially rescue cell death, although at higher [D-Ala] they still died (Fig. 5e), suggesting that ferroptosis is not the only form of oxidative cell death induced in the RPE1-hTERT-DAAO[TOM20] cells. In the current experimental setup, the induction of oxidative cell death occurs at levels of $H_2O_2$ production at the mitochondrial outer membrane that are high enough to be picked up by the HyPer7 sensor in the nucleus (Fig. 2), and hence outcompete the cytosolic antioxidant capacity. Collectively, our data suggest that mitochondria-derived $H_2O_2$ does not directly contribute to nuclear DNA damage because it does not reach the nucleus under (patho) physiological conditions. Even if sufficient $H_2O_2$ could be produced by mitochondria that it could reach and damage nuclear DNA, which seems unlikely based on the comparison with mitochondrial OCR (Fig. 1c), the cells die by oxidative cell death, preventing mutation accumulation and propagation (Fig. 6).

## Discussion

Mitochondrial respiration is a source of ROS and DNA can be oxidized by certain ROS (mainly ˙OH) in vitro[3,4]. The DDR is triggered in cells exposed to supraphysiological amounts of exogenous $H_2O_2$[20] and oxidized DNA bases, predominantly 8-oxo-dG, can be detected in patients with various diseases as well as in healthy subjects. 8-oxo-dG in the DNA can mispair with adenine, leading to C > A transversions after replication: a type of mutation that is enriched in several types of cancer[36]. These separate observations are often combined and extrapolated as if the (elevated) production and release of ROS (in the form of $H_2O_2$) by mitochondria can be directly translated to increased DNA oxidation and subsequent mutagenesis[11–13]. Based on this rationale, ROS derived from oxygen-dependent mitochondrial ATP production is frequently implicated in cancer initiation, progression, and treatment resistance, but also in aging in general. Unambiguous evidence for the *direct* oxidation and subsequent mutation of chromosomal DNA by ROS released from mitochondria at (patho) physiological levels, however, has been lacking.

Here we used a chemogenetic approach to induce local intracellular $H_2O_2$ production at the mitochondria or at nucleosomes to investigate the potential role of intracellular $H_2O_2$ production in DNA damage in untransformed human RPE1-hTERT cells. $H_2O_2$ generated at the nucleosomes indeed induced mutations and DNA strand breaks,

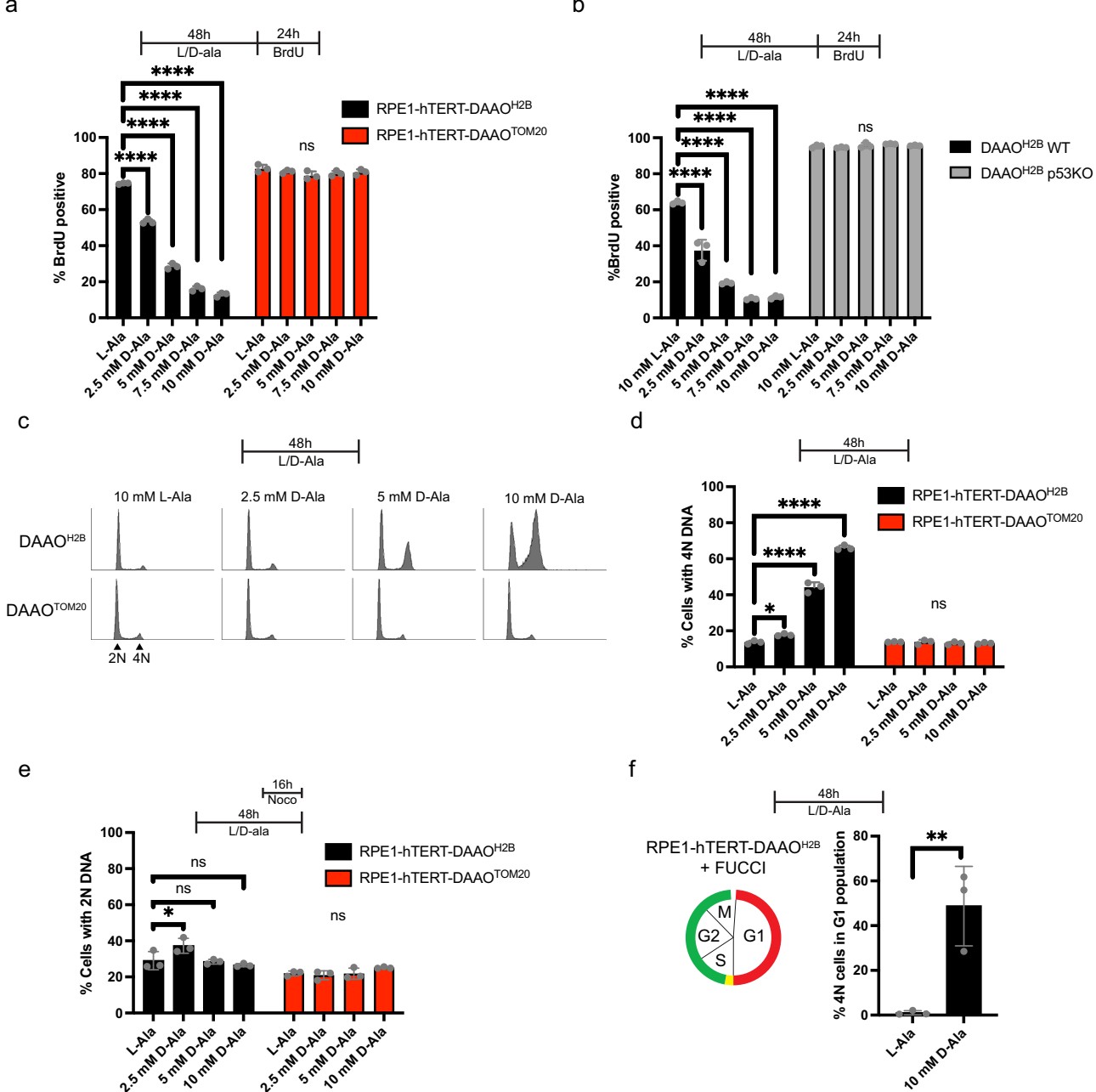

**Fig. 4 | Nuclear H$_2$O$_2$ production causes a tetraploid, p53-dependent cell cycle arrest, while mitochondrial H$_2$O$_2$ release does not. a** Quantification of the BrdU-incorporation assay. $N$ = 3 biological replicates -10.000 cells were analyzed per condition in each replicate. Data are presented as mean values -/+ SD. Statistical significance was determined by one-way ANOVA + Dunnett's multiple comparison test (****$p ≤ 0.0001$). Source data are provided in the Source Data file. **b** Same setup as in a, but comparing RPE1-hTERT-DAAO$^{H2B}$ WT cells with p53KO cells. $N$ = 3 biological replicates - 10.000 cells were analyzed in each replicate per condition. Data are presented as mean values -/+ SD. Statistical significance was determined by one-way ANOVA + Dunnett's multiple comparison test (****$p ≤ 0.0001$). Source data are provided in the Source Data file. **c** Cell cycle profile analysis of RPE1-hTERT-DAAO$^{H2B}$ and RPE1-hTERT-DAAO$^{TOM20}$ cells. **d** Quantification of (**c**), showing the percentage of cells with a 4 N DNA content. Dots represent 3 biological replicates - 10.000 cells were analyzed in each replicate per condition. Data are presented as mean values -/

+ SD. Statistical significance was determined by one-way ANOVA test + Dunnett's multiple comparison test (**$p ≤ 0.01$, ****$p ≤ 0.0001$, DAAO$^{H2B}$ 2.5 mM D-Ala = 0.0367, 5 mM D-Ala & 10 mM D-Ala ≤ 0.0001). Source data are provided in the Source Data file. **e** Quantification of cell cycle profiles of nocodazole treated cells. Dots represent 3 biological replicates - 10.000 cells were analyzed in each replicate per condition. Data are presented as mean values -/+ SD. Statistical significance was determined by one-way ANOVA test + Dunnett's multiple comparison test (*$p ≤ 0.05$, DAAO$^{H2B}$ 2.5 mM D-Ala = 0.0399). Source data are provided in the Source Data file. **f** Quantification of cell cycle profiles of RPE1-hTERT-DAAO$^{H2B}$ cells containing the FUCCI cell cycle indicator. Dots represent 3 biological replicates - 10.000 cells were analyzed per replicate per condition. Data are presented as mean -/+ SD. Statistical significance was determined by a two-sided unpaired $t$-test (**$p ≤ 0.01$, L-ala vs. D-Ala = 0.0097). Source data are provided in the Source Data file.

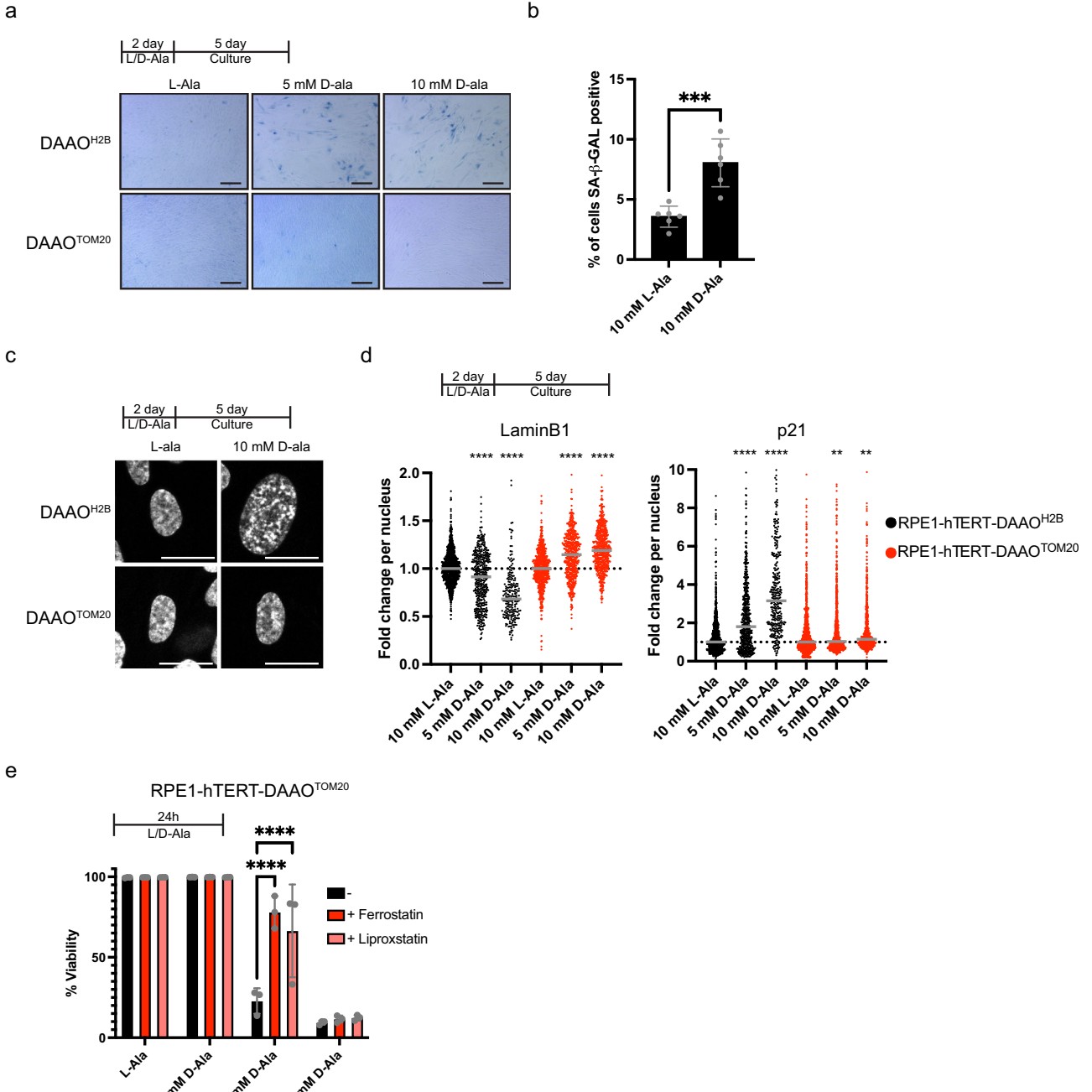

**Fig. 5 | Senescence and oxidative cell death are induced downstream of excessive localized H$_2$O$_2$ production. a** Senescence associated-β-galactosidase staining of RPE1-hTERT-DAAO$^{H2B}$ and RPE1-hTERT-DAAO$^{TOM20}$ cells. RPE1-hTERT-DAAO$^{H2B}$ cells become positive for the staining upon D-Ala addition. Staining is repeated 3 times, representative images are shown. Scale bar = 100 μm
**b** Quantification of Senescence Associated-β-galactosidase positivity of RPE1-hTERT-DAAO$^{H2B}$ cells treated wilt L-Ala or D-Ala. Experiment was repeated 3 times, 2 images of each were quantified. Data are presented as mean -/+ SD. Statistical significance was determined by a two-sided unpaired $t$-test (***$p \leq 0.001$, L-Ala vs. D-Ala = 0.0005). Source data are provided in the Source Data file. **c** Representative images of DAPI staining of RPE1-hTERT-DAAO$^{H2B}$ and RPE1-hTERT-DAAO$^{TOM20}$ cells. RPE1-hTERT-DAAO$^{H2B}$ cells undergo chromatin restructuring in response to D-Ala treatment, demonstrating a phenotype reminiscent of senescence-associated heterochromatin foci (SAHFs). Scale bar = 20 μm. Phenotype has been observed in at

least three independent experiments. **d** Quantification of IF stainings for senescence markers p21 and Lamin B1. Average signal per nucleus is shown. Induction of nuclear H$_2$O$_2$ results in a decrease in nuclear Lamin B1 signal and an increase in nuclear p21 intensity, both of which are markers of senescence. $n$ = 2 biological independent repeats. For each repeat 3 images were analyzed per condition. Horizontal line represents median. Two-way Anova + Dunett's multiple comparison test was performed (**$p \leq 0.01$, ****$p \leq 0.0001$, p21 DAAO$^{TOM20}$ 5 mM D-Ala = 0.0095, 10 mM D-ala = 0.0053). Source data are provided in the Source Data file.
**e** Quantification of cell viability by PI exclusion in RPE1-hTERT-DAAO$^{TOM20}$ cells. Dots represent 3 biological replicates. Per replicate, -10.000 cells were analyzed by FACS. Data are presents as mean -/+ SD. Massive cell death occurs at 15 mM D-Ala, which can be rescued by ferroptosis inhibitors ferrostatin and liproxstatin. Two-way Anova + Dunett's multiple comparison test was performed (****$p \leq 0.0001$). Source data are provided in the Source Data file.

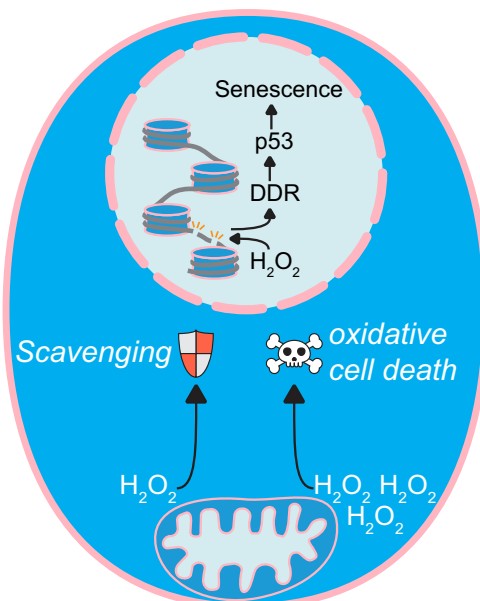

**Fig. 6 | Mitochondrial $H_2O_2$ release does not directly damage nuclear DNA.** Graphical abstract of our findings. Continuous elevated $H_2O_2$ levels in close vicinity to the DNA can cause mutations and DNA strand breaks, resulting in activation of the DNA damage response and ultimately senescence. In contrast, $H_2O_2$ released from mitochondria is not able to reach the nuclear DNA at physiological levels, and therefore does not directly contribute to nuclear DNA damage. Artificially high levels of mitochondrial $H_2O_2$ release, which is unlikely to be achieved in cells, does result in DNA strand breaks, but in parallel causes oxidative cell death.

resulting in activation of the DDR and subsequent p53-dependent senescence-like cell cycle arrest. Mitochondrial $H_2O_2$ release did not result in a detectable increase of $H_2O_2$ in the nucleus, even upon inhibition of the PRDX-TRX system, unless $H_2O_2$ was produced at levels that are one or two orders of magnitudes higher than what could likely be achievable physiologically. Furthermore, mitochondrial $H_2O_2$ release did not cause DNA strand breaks, activation of the DNA damage response or DNA mutations even at supraphysiological levels. Even at levels of mitochondrial $H_2O_2$ release that are equivalent to at least 10 to 100-fold the amount of ROS that is reportedly normally produced within mitochondria, cells maintain the capacity to proliferate without signs of DNA damage. Only when $H_2O_2$ production by DAAO^TOM20 at the mitochondria is boosted even further, so that it can be detected in the nucleus by Hyper7 (Supplementary Fig. 2B), it results in DNA strand breaks (Fig. 3c, d). However, and importantly, these levels of mitochondrial $H_2O_2$ production and release are unlikely to occur as a result of mitochondrial respiration and cells are not able to maintain redox homeostasis and die.

Although our data show that *direct* oxidation of chromosomal DNA by $H_2O_2$ released form mitochondria is unlikely, mitochondria-derived, cytoplasmic $H_2O_2$ could in principle indirectly result in DNA mutations by alternative modes of action. Firstly, $H_2O_2$-dependent and cysteine-thiol-based redox signaling has been shown to directly regulate DNA damage response and repair enzymes, as has been shown for ATM[37] and RAD51[28]. Therefore, mitochondrial $H_2O_2$ release could affect processes downstream of DNA damage stemming from other sources like replication stress. Secondly, free nucleotides, synthesized in the cytoplasm, could be oxidized by ˙OH formed from mitochondria-derived, cytoplasmic $H_2O_2$ in the Fenton reaction and subsequently incorporated in the DNA during S-phase. The presence of the enzyme MutT Homolog 1 (MTH1, also known as Nudix Hydrolase 1 (NUDT1)), which sanitizes the oxidized free nucleotide pool is an indication that this could be a feasible scenario[38]. However, we did not see an increase in DNA mutations by Whole Genome Sequencing of

D-Ala treated RPE1-hTERT- DAAO^TOM20 cells (Fig. 3e, f), which suggests that these two scenarios do not occur to a significant extent in our model system.

While superoxide is unlikely to be released from mitochondria due to efficient dismutation by SOD2 and SOD1, it is a relevant ROS because it can damage iron-sulfur (FeS) clusters, that are assembled in mitochondria[39,40]. Several proteins involved in DNA replication and DNA repair require FeS clusters as cofactors[41,42], and hence impaired FeS biosynthesis could exacerbate DNA damage stemming from other sources. Iron release could also drive the Fenton reaction mediated ˙OH production and subsequent damage. Intramitochondrial ROS could potentially damage the mitochondrial genome, which also has been suggested to contribute to cancer development[43]. It could therefore be that intramitochondrial ROS production indeed has different consequences as compared to mitochondrial $H_2O_2$ release. Our experiments targeting DAAO to the mitochondrial matrix and inter-membrane space suggest that intra-mitochondrial $H_2O_2$ production also does not lead to a DNA damage response (Supplementary Fig. 7) but leads to mitochondrial membrane depolarization and loss of respiration. The latter was not observed using DAAO^TOM20 (Supplementary Fig. 15). This indeed shows that intramitochondrial ROS production has effects besides mitochondrial $H_2O_2$ release. It will be interesting to see whether and how sustained (but viable) intramitochondrial ROS production affects genome integrity in the long term, possibly by mechanisms other than ROS release to the cytosol. Using KillerRed to produce superoxide directly in mitochondria instead of $H_2O_2$ could be a complementary approach to DAAO[44]. Qian et al.[45] used a mitochondria-targeted photosensitizer (MG-2I) to produce singlet oxygen upon exposure to near-infrared light in the mitochondrial matrix of HEK293T cells and showed that this led to a secondary wave of mitochondrial $H_2O_2$ release, which could be detected in the nucleus (by the first generation HyPer sensor) and led to damaged telomeres and subsequent ATM activation. We did not find evidence for ATM activation downstream of mitochondrial $H_2O_2$ release in RPE1-hTERT-DAAO^TOM20 cells regardless of p53 status (Fig. 3), nor in MCF7-DAAO^TOM20 breast cancer cells (Supplementary Fig. 9), at levels of $H_2O_2$ production in the physiological range and far above. Nuclear $H_2O_2$ was not detected in our experiments (Fig. 2), even though we used the more sensitive HyPer7 sensor. Mitochondria have also been suggested by Morita et al.[34] to be required for ATM activation in response to exogenous $H_2O_2$ in HepG2 cells, but in our model system we found no evidence for ATM activation by mitochondrial $H_2O_2$ (Fig. 3). These and other observations indicate that more work is needed to fully grasp the role of mitochondrial ROS in chromosomal DNA stability.

In this study we mostly used untransformed RPE1-hTERT cells to study the effects of local $H_2O_2$ production. However, genetic alterations like those obtained during cancer progression influence how cells respond to endogenously generated ROS. Genomic changes that lead to an increased reductive capacity and/or a loss of ferroptosis sensitivity[46,47] are also common adaptations found in cancer cells. Although we found no evidence for DNA damage downstream of mitochondria-derived $H_2O_2$ in (wild type p53) MCF7 breast cancer cells, we show that loss of p53 in RPE1-hTERT-DAAO^H2B cells leads to bypass of cell cycle arrest and senescence in response to $H_2O_2$ production at the nucleosome. In addition to genetics, the local metabolomic environment is also strongly linked to the redox properties of a cell[8,14,48], and can differ greatly between sites in the body or healthy and diseased state. For example, the tumor microenvironment generally provides less nutrients and oxygen, has a lower pH and contains more metabolic waste products[49], all of which could influence factors like $H_2O_2$ diffusion range and sensitivity to ferroptosis. It is therefore not unthinkable that mitochondria-derived ROS does play a role in the further accumulation of mutations during later stages of tumor progression.

Our findings have possible implications for cancer therapies that aim to boost ROS production to induce cancer cell senescence or death[50]. Our work indicates that ROS-inducing agents might indeed be useful to induce cell cycle arrest and senescence, but that these should be targeted to the nucleus and will likely only be effective in p53-expressing tumors. Oxidative cell death induced by cytosolic $H_2O_2$, like in RPE1-hTERT-DAAO$^{TOM20}$ cells seems to occur largely independent of p53.

As mentioned, 8-oxo-dG has been detected in patient material, but the subcellular source of the oxidant is not known. Based on our observations we envision two scenarios how endogenous ROS could lead to accumulation of 8-oxo-dG in chromosomal DNA. (1) Guanine in the nuclear genomic DNA is subject to oxidation, which would mean that $H_2O_2$ is produced and reacts to form $^{\cdot}OH$ in close proximity to the DNA. A potential source of $H_2O_2$ in the nucleus is for instance Lysin specific demethylase 1 (LSD1), which removes mono- and dimethyl groups on lysine 4 of histone 3 (H3), thereby also generating $H_2O_2$[51] and has indeed been shown to lead to the local formation of 8-oxo-dG[52]. NOX4[53] and the oxidoreductase MICAL1[54] have also been shown to produce $H_2O_2$ in the nucleus. (2) Free deoxy-GTP nucleotides are oxidized in the cytoplasm followed by incorporation into the nuclear DNA. Which of these two scenarios takes place could in principle be distinguished, because free nucleotide incorporation in the DNA gives rise to a different SBS signature compared to direct 8-oxo-dG formation in nuclear DNA[55,56]. We will extend our Whole Genome Sequencing experiments (Fig. 3e, f) and analysis in future studies to understand where in the genome these mutations occur, at what bases (Guanine only or also others) and in what trinucleotide contexts (COSMIC mutational signatures[57]).

Our experiments were carried out under atmospheric oxygen tension, which has been shown to lead to higher mutational rates stemming from unrepaired 8-oxodG in cells in culture[57]. The source of the oxidant that induces these culture-induced mutations remains elusive, but our data presented here suggest that mitochondrial respiration is not directly involved. Indeed, our sequencing data show that several hundred background single base substitutions can be found when comparing cells at the start of control L-Ala treatment with monoclonals grown out after 4 rounds of treatment and 3 days recovery, which may be due to oxidative modification. It may be interesting to see whether and how this number changes in case the experiment would be carried out under a more physiological oxygen tension, but since DAAO activity also depends on oxygen it will be difficult to estimate the rates of $H_2O_2$ production under these circumstances.

Recent studies in yeast, plant and human cells concluded that release and diffusion of $H_2O_2$ from mitochondria is very limited due to highly efficient scavenging by abundant peroxidases[14–17,58]. But these studies and many others including from our laboratory[17] have shown that mitochondrial ROS does play a vital role in cellular signaling. An open question remains at what subcellular localization these signaling events are initiated, whether the localization determines the downstream phenotype and how these signals are transduced.

Induction of the ATM/ATR dependent DDR and a subsequent p53-induced cell cycle senescence-like arrest observed upon DAAO activity at the nucleosome is reminiscent of the response of RPE1 and other cell types to a bolus of exogenous $H_2O_2$[59,60]. This would suggest that exogenously added $H_2O_2$, unlike $H_2O_2$ produced at the mitochondria in RPE1-hTERT-DAAO$^{TOM20}$ cells does make it into the nucleus to induce DNA damage. This apparent discrepancy could potentially be explained by the dose of exogenous $H_2O_2$ that is used in these studies to induce senescence, which was 150–250 $\mu M$: much higher than what can be achieved by localized DAAO. Bolus addition of these amounts of $H_2O_2$ lead to overoxidation of PRDX (Supplementary Fig. 1), which may be why it can induce DNA damage. On the other hand, we show that DNA damage in D-Ala treated RPE1-hTERT-DAAO$^{H2B}$ cells was triggered in the absence of overoxidized PRDX. What remains to be elucidated is why these high levels of exogenous $H_2O_2$ do not induce widespread oxidative cell death like we observed in response to prolonged exposure of RPE1-hTERT-DAAO$^{TOM20}$ cells to >20 mM D-Ala, which does not lead to PRDX overoxidation. Overoxidation of PRDX has been shown to protect yeast cells from cell death upon bolus $H_2O_2$ treatment, and it could be that similar mechanisms underlie our observations[60].

Senescence markers have been shown to be heterogeneous and may depend on the type of senescence induction. Further studies are needed to better classify whether the observed arrest in our RPE1-hTERT-DAAO$^{H2B}$ cells also display other markers like a Senescence Associated Secretory Phenotype. The oxidative cell death induced in our RPE1-hTERT-DAAO$^{TOM20}$ could be partially blocked by ferroptosis inhibitors Ferrostatin and Liproxstatin, but only with a narrow window of [D-Ala]. We therefore think it is premature to state the observed oxidative cell death at artificially high mitochondrial $H_2O_2$ release is truly and only ferroptotic cell death. We do think that our model system may be useful in future work for elucidating mechanisms of ferroptosis. It can be used to study from what subcellular localizations ferroptosis can be triggered, whether there is involvement of mitochondrial lipid peroxidation and whether cells survive or die by other means when ferroptosis is blocked.

Taken together, our work shows that not only the dose but also the localization and kinetics of $H_2O_2$ production can have dramatic effects on phenotypic outcome. We conclude that, at physiological levels, the release of ROS by mitochondria is not a significant source of direct DNA oxidation or indirect DNA mutation and is therefore not likely to contribute considerably to cancer initiation and aging.

## Methods

### Cell culture and treatment

Retinal pigment epithelial cells (hTERT RPE-1, ATCC, CRL-4000) were cultured in DMEM-F12 (Gibco, 11320033), supplemented with 10% FBS (Bodinco), 100 Units Penicillin-Streptomycin (Sigma-Aldrich, P4458) and 2 mM L-glutamine (Sigma-Aldrich, G7513). Cells were grown at 37 °C under 6% $CO_2$ atmosphere. Human breast cancer cells (MCF7, ATCC, HTB-22) and human epithelial kidney cells (HEK293T, ATCC, CRL-3216) used for lentiviral production were cultured in DMEM-High Glucose (Sigma-Aldrich, DB6429) supplemented with 10% FBS, 100 Units Penicllin-Streptomycin and 2 mM L-glutamine. L-alanine (Sigma-Aldrich, A7626) and D-alanine (Sigma-Aldrich, A7377) were dissolved in PBS to create a 1 M stock solution and frozen in aliquots, only to be thawed once. Addition of alanine was always precluded by refreshing medium. For trapping cells in mitosis to visualize a $G_{0/1}$ arrest, 250 ng/ml nocodazole (Sigma-Aldrich, M1404) was used. For blocking ferroptosis, 500 nM ferrostatin (Sigma-Aldrich, SML0583) or 200 nM Liproxstatin (Sigma-Aldrich, SML1414) was added 1 h before treatment with L- or D-Ala. For inhibiting thioredoxin reductase, 1 $\mu M$ Auranofin (Sigma-Aldrich, A6733) was used.

### Cloning, lentiviral infection and Cas9 KO procedure

The original genetic sequence for DAAO from *Rhodotorula gracilis* was codon optimized for human cells using the IDT codon optimization tool. The last 3 amino acids containing the localization signal were excluded and the sequence was ordered as a geneblock (Integrated DNA technologies). This construct was a kind gift from Dr. Lucas Bruurs and described before[17]. The genetic sequence for H2B-mScarlet-I-DAAO and TOM20-mScarlet-I-DAAO was introduced by infusion cloning (Takara Bio) into a lentiviral backbone under control of a human EF-1α promoter and containing a puromycin selection cassette. Plasmids containing the genetic sequence for NLS-HyPer7 and NES-HyPer7 were a kind gift from Dr. Vsevolod Belousov. These sequences were cloned (by infusion cloning) into a lentiviral backbone under control of a human EF-1α promoter and containing a blasticidin selection cassette. For virus production, lentiviral DAAO and HyPer7

constructs were transiently transfected into HEK293T cells together with 3rd generation packaging vectors using PEI Max (Tebubio, 24765-1). Virus containing medium was filtered using a 0.45 μm filter and virus was further purified by ultracentrifugation. Virus was added to RPE1-hTERT, RPE1-hTERT-FUCCI (described in[33]) and MCF7 cells together with 8 μg/ml polybrene (Sigma-Aldrich, H9268). Cells were selected using 5 μg/ml puromycin (Santa Cruz) and 10 μg/ml blasticidin (InvivoGen) for RPE1-hTERT cells and 1 μg/ml puromycin for MCF7 cells. Single cells were expanded in 96-wells plates and expanded to create monoclonal lines. For the HyPer7 lines, only cells with high HyPer7 expression were sorted.

PCR reactions for cloning were performed using Q5 High-Fidelity DNA polymerase (New England Biolabs, M0491), according to manufacturer's protocol. DNA and plasmid isolation from agarose gels, PCR reactions or bacterial culture was performed using QIAquick Gel extraction kit (Qiagen, 28706), QIAprep spin miniprep kit (Qiagen, 27104) and Maxiprep purification kit (LabNed, LN2400007).

The construct coding for Cas9, a p53-targeted guide RNA (5' CGCTATCTGAGCAGCGCTCA 3') and a GFP expression reporter was made using the PX458 plasmid and method described by Ran et al.[61]. RPE1-hTERT-DAAO^H2B and RPE1-hTERT-DAAO^TOM20 were transiently transfected using PEI Max. After 48 h GFP positive cells were single-cell sorted into a 96-wells plate using FACS. P53 knock-out status of clones was confirmed by testing sensitivity for the p53 stabilizer Nutlin-3a (10 μM for 16 h, Cayman Chemical, 10004372) and Western Blotting for p53 protein.

## Antibodies
Antibodies used for Western Blot were pChk1 S345 (Cell Signaling, CS2348, lot# 18, 1:1000), Chk1 (Santa Cruz, SC8408, lot# A1713, 1:1000), pChk2 T68 (Cell Signaling, CS2661, lot# 13, 1:1000), Chk2 (Cell Signaling, CS3440, lot# 4, 1:1000), γH2AX (Millipore, 05-636, lot# 3824772, 1:1000), p53 (Santa Cruz, SC-126, lot# E2521, 1:2000), p21 (BD Biosciences, 556430, lot# 1173681, 1:1000), ATM (Abcam, ab78, lot# GR3286342-11, 1:2500) pATM S1981 (Abcam, ab81292, 1:2500) tubulin (Millipore, CP06 OS, lot#3239856 1:2000), PRDX2 (Abcam, ab109367, lot# 1000538-1 1:2000) and PRDXSO$_{2/3}$ (Abcam, ab16830, lot# GR3294252-3, 1:1000).

Antibodies used for IF were LaminB1 (Merck, ZRB1143, lot# Q3250182, 1:500) and p21 (Merck, ZRB1141, lot# 3307439, 1:800).

## Live imaging of DAAO localization & Immunofluorescence
Live cell imaging was performed to check whether the mScarlet-I-DAAO fusion proteins localized to the correct subcellular compartments. Cells were grown on glass-bottom dishes (WillcoWells, GWST-3522). Half an hour prior to imaging, medium was replaced with medium containing 2 μg/ml Hoechst33342 (Thermo Fischer Scientific, 62249) and 200 nM MitoTracker Green (Invitrogen, M7514). After half an hour, medium was replaced with fresh medium. Cells were imaged on a confocal LSM880 microscope (Zeiss), using ZEN 2.3 Sp1 (version 14.0.20.201). Hoechst was excited at 405 nm and emission measured at 415–481 nm, Mitotracker Green was excited at 488 nm and emission measured at 499–545 nm and mScarlet-I was excited at 568 nm and measured at 571–677 nm. Z-stacks were acquired and displayed as maximum projections using FIJI (version 2.9.0/1.54b) software[62].

For immunofluorescence staining of senescence markers, cells were grown in 8-well Lab-Tek slides with 1.5 borosilicate glass bottom (Thermo Fischer Scientific, 155409). Cells were treated for 2 days with L- or D-Alanine and kept in culture for an additional 5 days. Cells were then washed on ice twice with PBS, after which they were fixed with 4% PFA (diluted in PBS from 16% stock, Electron Microscopy Sciences, 15710) at room temperature for 15 min. Cells were then washed twice with PBS and quenched with 50 mM Glycine (in PBS, Sigma-Aldrich, W328712) for 10 min. Then, cells were twice washed with PBS and permeabilized with 0.1% TX-100 (in PBS, Millipore, 1.08603). Cells were

washed an additional 2 times, followed by blocking with 2% BSA solution (Sigma-Aldrich, A9647) in PBS. Primary antibody staining was performed overnight at 4 °C in 2% BSA PBS solution containing normal goat IgG (1:10000, Santa-Cruz, 2028). Cells were washed 3x and secondary staining with alexa488 and alexa647 conjugated antibodies (1:500, Life Technologies) was performed at room temp for 1 h together with DNA staining by DAPI (1 μg/ml, Sigma-Aldrich, D9564). Cells were washed an additional 3x and imaged on an LSM880 confocal microscope (Zeiss). Alexa488 was excited at 488 nm and emission measured at 493–558 nm and Alexa647 was excited at 633 nm and emission measured at 638–744 nm. Mean signal per nucleus was determined using FIJI (version 2.9.0/1.54b) software. Nuclei were segmented based on DAPI staining using the StarDist plugin[63]. Mean signal of segmented nuclei was measured.

## Measuring DAAO activity by oxygen consumption rate and mito stress test
The reaction catalyzed by DAAO consumes an equimolar amount of molecular oxygen per molecule of H$_2$O$_2$ produced. Changes in Oxygen Consumption Rate (OCR), which can be measured for instance in a Seahorse XFe24 analyzer (Agilent), can therefore be used to estimate DAAO activity upon addition of D-Ala[23]. Briefly, 24-well V7 Seahorse culture plates (Agilent, 100777) were coated with 50 μl of 50 μg/ml rat tail collagen (dissolved in 0.1% acetic acid, Corning, 354236) for 20 min at RT, and subsequently washed with PBS. 4*10^4 DAAO-expressing RPE1-hTERT cells were seeded per well. 4 wells remained unseeded for background correction. A prewarmed Seahorse XFe24 analyzer (Agilent) with Wave (version 2.6.1.56) software was used to measure OCR. On the day of the assay, the medium of the cells was replaced with XF base medium (Agilent, 103334) supplemented with 2 mM L-Glutamine (Sigma-Aldrich, G7513), 17.5 mM glucose (Gibco, A2494001), 1 mM Na-Pyruvate (Gibco, 11360070), and 0.5 mM NaOH. Oligomycin A (final concentration 2 μM, Cayman Chemical, 20185) and L/D-Alanine was added to injection ports. OCR was normalized to the third measurement after oligomycin addition. For MCF7 cells, 24-well V7 Seahorse culture plates were coated with fibronectin (homemade) diluted 1:100 in PBS for 1 h at 37 °C. Per well 2*10^4 cells were plated. XF base medium was supplemented with 2 mM L-Glutamine (Sigma-Aldrich, G7513), 25 mM glucose (Gibco, A2494001) and 0.5 mM NaOH. As MCF7 cells slowly recover OCR after oligomycin addition, WT MCF7 cells were taken along to correct for the increase in OCR due to oligomycin recovery. A more detailed protocol on measuring DAAO activity by oxygen consumption can be found here[23].

To measure mitochondrial function, a mito stress test was performed by injection of 2 μM oligomycin A (Cayman Chemical, 20185), 1 μM FCCP (Sigma-Aldrich, C2920), 1 μM Antimycin A (Sigma-Aldrich, A8674) and 1 μM Rotenone (Sigma-Aldrich R8875). After measuring, cells in each well were lysed using a buffer containing 0.1% TX-100, 10 mM Tris pH 7.5, 5 mM EDTA and 100 mM NaCl. Plates were put at −20 °C overnight and protein concentration per well was measured by use of a Pierce BCA protein assay kit (Thermo Fischer Scientific, 23225) according to manufacturer's protocol. OCR measurements were then normalized by protein concentration.

## HyPer7 and FUCCI measurements
RPE1-hTERT cells stably expressing HyPer7 were plated in ibiTreated 8-well μ-slides (Ibidi). Measurements were performed on a Cell Observer microscope (Zeiss) using a 10 x magnification and ZEN 2.6 blue software (v.2.6.76.00000). Cells were excited at 385 nm and 475 nm and emission was measured using a 514/44 BP bandpass filter every 10 min for up to 48 h. After one round of imaging, L/D-Ala was added from 10 x concentrated working solutions in PBS to ensure rapid diffusion. Data was quantified using FIJI (version 2.9.0/1.54b) software[62]. Background of both channels was subtracted in FIJI using the rolling ball algorithm, and channels were thresholded to remove

background pixels (NaN). Ratiometric movies of $475_{ex}/385_{ex}$ were made using the image calculator, and mean pixel intensity was measured for each timepoint. The $475_{ex}/385_{ex}$ ratio was transformed into a percentage of HyPer7 oxidation by setting the lowest ratio measured to 0% and the highest ratio measured after exogenous $H_2O_2$ addition to 100%. Cell cycle analysis by live imaging of FUCCI expressing cells was performed as described previously[33]. Red-Green Color-blind friendly false color images were prepared with mKO2-CDT1 ($G_{0/1}$) displayed in Blue and mAG1-Geminin (S/$G_2$) in Green.

## SDS-PAGE and western blot

Protein lysates were obtained by washing cells with PBS and subsequently lysed and scraped in 1 x Sample Buffer (2% SDS, 5% 2-mercaptoethanol, 10% glycerol, 0.002% bromophenol blue, 300 mM Tris-HCl pH 6.8,). Samples were first heated for 5 min at 95 °C before running SDS-PAGE and standard blotting procedure. Samples for γH2AX staining were transferred onto a 0.2 μm nitrocellulose membrane (Cytvia Amersham, 10600001), whereas other samples were transferred onto a polyvinylidene difluoride (PVDF) membrane (Millipore, IPVH00010). Samples were blocked for 1 h at 4 °C in TBS-Tween (1% v/v), containing 1% BSA (Sigma-Aldrich, A9647). Primary antibody incubation was performed overnight or over weekend at 4 °C in TBS-T containing 1% BSA. Secondary antibodies conjugated with horseradish peroxidase (1:10000, Rockland, 610-4302 and 611-1302) in TBS-Tween were incubated for 1 h at room temperature. Blots were imaged on Image Quant LAS (GE HealthCare) with ImageQuant LAS4000 (version 1.3) software. Contrast was adjusted for clarity by linear image processing in Adobe Photoshop if required. Uncropped Western Blots can be found in the associated Source Data file.

For non-reducing SDS-PAGE, cells were incubated for 5 min with prewarmed PBS containing 12.5 mg/ml N-Ethylmaleimide (Millipore, E3876) at 37 °C prior to harvesting to fix the cysteine-dependent redox state. Cells were then washed with PBS and lysed in buffer containing 50 mM Tris-HCl (pH 7.5), 0.1% TritonX-100 (Millipore, 1.08603), 2.25 mM $MgCl_2$ (Sigma-Aldrich, 8.14733), 0.1 M NaCl (Supelco, 1.06404), NaF (Supelco, 106449), 0.1% Aprotinin (Sigma-Aldrich, A1153), 0.1% Leupeptin (Sigma-Aldrich, L2023) and 100 mM iodoacetamide (Sigma-Aldrich, I1149). Samples were divided over two tubes and a reducing and a non-reducing sample was prepared by adding SDS sample buffer with or without beta-mercaptoethanol.

## Flow cytometry

Cell viability was assessed by propidium iodide (PI, Sigma-Aldrich, P4170) exclusion. Medium and subsequent wash with PBS were collected, after which cells were trypsinized, resuspended and added to the collected medium + PBS (15 ml falcon tubes), spun down (1500 RPM 5 min, Beckmann tabletop centrifuge with swingout rotor) and put on ice. Supernatant was removed, and the pelleted cells were resuspended in PBS containing 20 μg/ml propium iodide.

For DNA profiles, cells were collected in a similar fashion. After centrifugation and removal of supernatant, 200 μl of PBS was added to prevent cells from aggregating during fixation. Cells were fixed by dropwise addition of 5 ml of ice-cold 70% ethanol while vortexing. Samples were left at 4 °C overnight or longer. On the day of measurement, fixed cells were pelleted (1500 RPM 5 min) and resuspended in PBS containing 20 μg/ml propium iodide and 200 μg/ml RNase (Millipore, 70856). In case of cells containing the FUCCI cell cycle reporter, live cells were collected, pelleted (1200 RPM 5 min) and resuspended in PBS containing 2 μg/ml of the membrane permeable DNA staining Hoechst33342 (Thermo Fischer Scientific, 62249) (Thermo Fischer Scientific).

For BrdU measurements, cells were cultured in medium containing 10 μM BrdU (Invitrogen, 000103) 24 h prior to collecting cells. Cells were collected and fixed in the same fashion as for DNA profiles. On the day of measurement, anti-BrdU staining was performed. First, after

centrifugation the cell pellet was resuspended in 0.1 M HCL solution containing 0.5 mg/ml of the protease pepsin (Sigma-Aldrich, 1.07185) and left for 20 min at room temperature. After washing with PBS containing 0.1% BSA and 0.5% Tween, cells were pelleted and treated with 2 M HCL for 12 min at 37 °C to denature the DNA, after which double the volume of borate buffer (0.1 M $H_3BO_3$ set at pH 8.5 with 0.1 M $Na_2B_4O_7$) was added. After washing and centrifugation, the cell pellet was incubated with PBS/BSA/Tween containing BrdU-FITC antibody (1:50, BD Biosciences, 347583) for 1 h on ice in the dark. Cells were centrifuged and resuspended in PBS/BSA containing 20 μg/ml PI and 200 μg/ml RNase.

To analyze mitochondrial membrane potential, cells were cultured in medium containing 100 nM tetramethylrodamine (TMRM, Thermo Fischer Scientific, T668) for 30 min at 37 °C. Cells were collected in 15 ml tubes, pelleted (1200 RPM 5 min) and resuspended in 250 μl PBS. As a positive control for loss of membrane potential, 10 μM FCCP was added for at least 15 min prior to measuring.

Flow cytometry samples were measured on BD FACSCelesta Cell analyzer (BD biosciences) and analyzed with BD FacsDiva Software. Gating strategies for Flow Cytometry experiments can be found in Supplementary Fig. 16.

## Crystal violet staining

Cells were plated in 6-well plates. After 1 day, cells were treated with L- or D-Alanine for 24 h. Cells were then washed 2x with PBS to remove dead cells and fixed with ice-cold (−20 °C) methanol for 5 min. Methanol was removed and replaced with crystal violet solution (0.5% w/v in 25% v/v methanol in $H_2O$) for 10 min. Crystal violet was removed and wells are washed with water and airdried overnight. Plates were imaged on an Epson 3170 photo scanner (Epson). For quantification, the staining was dissolved in 10% acetic acid for 30 min, transferred to a 96-wells plate and measured in a plate reader by spectrophotometry at 595 nm.

## Senescence-associated β-galactosidase staining

For the SA-β-gal staining, cells were cultured in 6-well plates. Cells were treated for 2 days with L- or D-Ala and kept in culture for an additional 5 days. Then cells were washed twice with PBS and fixed for 15 min with 3.7% formalin (37% formalin diluted in PBS (SAFC, 1.04002). Cells were washed twice with PBS and then incubated with freshly made SA-β-gal buffer containing 25 mM $Na_2HPO_4$ (Supelco, 1.06586), 7.4 mM Citric Acid (SAFC, 1.00241), 150 mM NaCl (Supelco, 1.06404), 2 mM $MgCl_2$ (Sigma-Aldrich, 8.14733), 5 mM Potassium Ferricyanide (Sigma-Aldrich, 702587), 5 mM Potassium Ferrocyanide (Sigma-Aldrich, 60279) and 0.1% X-Gal (from a 4% X-Gal solution in dimethylformamide, Sigma-Aldrich, 71077). Buffer was passed through a 0.2 micron syringe filter before X-Gal was added. Cells were incubated overnight at 37 °C, at ambient $CO_2$. Cells were rinsed 2x with PBS and kept under 70% ethanol. Cells were imaged on an EVOS-M5000 microscope (Thermo Fisher).

## Comet assay

Alkaline comet assays were performed using the CometAssay single cell gel electrophoresis assay kit (R&D systems 4250-050-K) according to the manufacturers' instructions. Cells were treated for 2 h with L/D-Ala after which they were trypsinized and counted to make $10^5$ cells/ml dilutions. Cell lysis was performed overnight, DNA alkaline unwinding for 1 h and electrophoresis for 1 h, all at 4 °C. DNA staining was performed using Midori Green Advance for 30 min at room temperature (1:5000 in TE buffer pH7.5, Nippon Genetics, MG04). Comets were imaged on an EVOS-M5000 microscope (Thermo Fisher) using the GFP filter block. Comets were manually analyzed using FIJI (version 2.9.0/1.54b) software. Samples were blinded before quantification. The area and mean pixel value of the comet head and the complete comet were measured. In addition, the mean pixel value of the local background

was determined. The mean pixel value of the background was subtracted of the mean pixel value of the comet head and total comet. These values were then multiplied with the area of the comet head or total comet to create the integrated densities corrected for background signal. The integrated density of the comet tail value was calculated by subtracting the integrated density of the comet head from the integrated density of the total comet. The percentage DNA in comet tail was then calculated by dividing the integrated density of the comet tail value with the integrated density of the total comet. Per condition 50 comets were analyzed.

**Whole Genome Sequencing (WGS) data for mutational analysis**
RPE1-hTERT-DAAO$^{H2B}$; p53KO and RPE1-hTERT-DAAO$^{TOM20}$; p53KO cells were treated for 2 h with 20 mM L-Ala or D-Ala followed by washing and 22 hrs recovery for 4 consecutive days to achieve high levels of $H_2O_2$ production without inducing cell death. Cells were then cultured for an additional 3 days to allow DNA modifications to be translated into mutations and subsequently single cell sorted to allow for monoclonal outgrowth. Once clones obtained 70–80% confluency in a 10 cm culture dish, genomic DNA was harvested using the DNeasy blood and tissue kit (Qiagen, 69504), according to manufacturer's protocol. DNA was also obtained of both lines prior to L-Ala and D-Ala treatment to control for pre-existing mutations ($T_0$). WGS libraries were generated using standard protocol (Illumina), which were sequenced to 15 x coverage (2 ×150 bp) on an illumina Novaseq 6000 machine. WGS read alignment, variant calling, and filtering, was done as described before[64]. Briefly, the human reference genome (GRCh38) was used to map the sequencing reads. Reads were marked for duplicates and realigned using the Genome Analysis Toolkit (GATK) (v4.1.3.0). Raw variants versus $T_0$ were called in multi-sample mode using the GATK HaplotypeCaller and GATK-Queue with default settings and the option 'EMIT_ALL_CONFIDENT_SITES'. Quality of variants and reference positions was determined using GATK VariantFiltration. A description of the complete data analysis pipeline is available at https://github.com/ToolsVanBox/NF-IAP (v1.3.0).

**Statistics and reproducibility**
Statistical analyses were performed using GraphPad Prism (version 10.1.1). Details on the amount of replicates, type of test and exact p-values are indicated in the figure legends. All findings reported here were observed multiple independent times, as stated in the figure legends. No statistical methods were used to predetermine sample size. No data were excluded from the analyses. For experiments where analysis was performed manually (e.g. comet assay), investigators were blinded to the sample's treatment conditions during analysis.

**Reporting summary**
Further information on research design is available in the Nature Portfolio Reporting Summary linked to this article.

## Data availability
Whole Genome Sequencing data is available at the European Nucleotide Archive (https://www.ebi.ac.uk/ena/browser/home) with the accession code PRJEB72014. The raw data generated in this study to plot graphs and uncropped Western blots are provided as Source Data file. Source data are provided with this paper.

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

## Acknowledgements

We would like to thank Tessa Vreeman for help with constructing lentiviral HyPer7 plasmids. We are grateful to dr. Peter de Keizer for advice on the assessment of senescence, Livio Kleij for technical support with the HyPer7 measurements, dr. Fried Zwartkruis for critical reading of the manuscript and our colleagues at the Center for Molecular Medicine, UMC Utrecht for their valuable input and suggestions. This work was funded by grants from the Dutch Cancer Society (KWF Kankerbestrijding, KWF grant 14798 and KWF UU2014-6902) to TBD. BMTB and RvB are members of the Oncode Institute, which is partly funded by the Dutch Cancer Society (KWF Kankerbestrijding).

## Author contributions

D.M.K.v.S., P.E.P., W.T.f.d.T., J.P.K. and T.B.D. designed, performed and analysed experiments. M.J.v.R. performed bioinformatic mutational analysis. T.M.F.L. and J.L. constructed, validated and analysed the monoclonal MCF7-DAAO lines. S.Z. made the CRISPR/Cas9 knockout p53 KO lines, R.v.B. designed and supervised the sequencing and mutation analysis experiments. B.M.T.B. provided critical input on the experimental setup and the manuscript. D.M.K.v.S., S.d.H. and T.B.D. conceived the study, D.M.K.v.S. and T.B.D. wrote the paper.

## Competing interests

The authors declare no competing interests.
