## [Peer Review File · Nature Communications]

Mitochondrial H₂O₂ release does not directly cause damage to chromosomal DNAREVIEWER COMMENTS

Reviewer #1 (Remarks to the Author):

DNA damage is often associated with an overabundance of reactive oxygen species (ROS). Since mitochondria are one of the main sources of ROS they have been frequently implied in the induction of DNA damage through the release of ROS, specifically H₂O₂. The authors of this study tackled this hypothesis by genetically engineering cells and equipping them with a D-amino acid oxidase (DAAO) localized either to the nucleus or the cytosolic side of the mitochondrial outer membrane (TOM20-DAA). Upon addition of D-alanine, DAAO generates H₂O₂. The authors found that hardly any H₂O₂ reaches the nucleus when produced by TOM20-DAAO although DAAOs at both locations produced equal amounts of H₂O₂. As a consequence, there was also no DNA damage, senescence etc. observable. Upon production of extremely high H₂O₂ concentrations at mitochondria, cells were driven to ferroptosis emphasizing this already known mitochondria-ferroptosis axis.

This is a well-conceived and technically sound study that addresses an important question in the field, namely to which extent does the release of mitochondrial H₂O₂ contribute to cellular damage. The authors demonstrate that hardly any H₂O₂ reaches the nucleus when produced by TOM20-DAAO. Based on this main finding the authors can also convincingly demonstrate that NO H₂O₂ also does not lead to DNA damage and does not pose any danger to the cell (essentially all experiments following Figure 2). However, it is already known from previous studies that only minute amounts of H₂O₂ can diffuse away from mitochondria upon its release. This is due to the strong antioxidative systems of the cytosol. Some of these previous findings have even been made employing exactly the same biosensor as used by the authors (e.g. Hohne et al, EMBO J 2022; Koren et al, bioRxiv 2023; de Cubas et al, Antioxidants 2021). From these previous findings and the present study, it is therefore not completely surprising that mitochondrial H₂O₂ does not lead to nuclear DNA damage in healthy cells as even the high D-Ala concentrations used in this study (10-20 mM) lead to nuclear H₂O₂ levels (produced by TOM20-DAAO) that are well below what lead to any damage with nuclear DAAO.

I have one major concern with this study: I am not sure whether this chemo-genetic approach in "healthy" cells really reflects pathological conditions under which NADPH supply might fail (which ultimately drives antioxidative responses in the cytosol and has been shown to allow H₂O₂ to travel much further!) or mitochondria start localizing more perinuclear. Thus, I am also not sure whether with this study any contribution of mitochondrial H₂O₂ to nuclear DNA damage can be excluded.

Minor points:

- (1) further Markers for Senescence should be used e.g. Senescence-Associated Secretory Phenotype (SASP)
- (2) lipid peroxidation at mitochondria should be looked as potential driver of ferroptosis.

Reviewer #2 (Remarks to the Author):

It is well known that ROS can induce DNA damage, stimulate the DDR and contributes to cellular stress and mutagenesis. Given that mitochondrial respiration is a major contributor to cellular ROS, it was assumed that mitochondrial derived ROS is the central contributor to nuclear DNA damage. However, there was a lack of direct evidence supporting this claim. The authors have set up a clever system to directly test this model. They sought to determine whether mitochondrial-derived ROS influences nuclear DNA and cellular responses to DNA damage. Ectopic expression of DAAO, either at the mitochondrial outer membrane (TOM20) or nuclear histones (H2B), allows for inducible ROS production directly in the nucleus or at mitochondria. The authors then used a combination of imaging, flow cytometry and biochemical assays to determine cellular responses to nuclear and mitochondrial derived ROS. The authors central finding is H₂O₂ generated at mitochondria does not affect dsDNA

breaks, cell cycle arrest, DNA damage (comet tails), nor cell death (at lower concentrations). These findings are quite surprising. We think that this is an important study and because the authors are going against dogma, we feel that additional experiments are required to shore up the central premise. Overall, we are excited about this work, and feel that these additional experiments would provide better support to their conclusion that mitochondrially derived H₂O₂ does not cause genomic DNA damage.

Major comments:

-A major concern is the inconsistency of time points for the assay, and correlating the findings at one time point with findings at another time point. For example, the timing of H₂O₂ production was performed at 9 hours, prdx2 at 2 hours, cell death assays were performed at 24h, DNA damage at 48 hr. It would be ideal if the authors examined H₂O₂ production closer to the time of the cell behavior? Does the H₂O₂ generation stay high? Does it go down, then how does that relate to cell cycle/DNA damage/etc?

-Due to these surprising (but very interesting) results, it is important to show universality, and we suggest performing key experiments (H₂O₂ production at mitos or nucleus, and examining DNA damage) in at least one other non-transformed cell line.

-Most of the experiments in the supplemental data appear to have only been performed once. Please indicate biological and technical replicates, and statistics. This is also true for Figure 5a-c.

-Given that this paper is all about the localization of the ROS production, it is important to observe another ROS biosensor to visualize levels (for example, quantification of MitoSOX, or mitochondrially-localized HyPER).

-The authors are also encouraged to use another cell viability assay that provides more quantitative data. There are several flow-based assays out there, or even direct cell counts.

Minor Comments:

-In text typo: 'Reviewed in PMID 20064600'

-'Treatment of cells with a high dose of exogenous H₂O₂ has been shown to activate both ataxia telangiectasia and RAD3-related protein (ATR) and ataxia telangiectasia mutated protein (ATM) dependent DNA damage response (DDR), indicative of single and double strand DNA breaks respectively.' – reference needed

In text typo: 'Therefore, we monitored whether the DDR was activated in RPE1-hTert-DAAOTOM20 and RPE1-HTERT-DAAOH2B lines upon D-Ala treatment in.'

A better explanation for how the D-ala-induced ROS production relates to normal levels of ROS is needed. Its hard to interpret what these data mean without understanding this key point, and the text did not make it clear.

Reviewer #3 (Remarks to the Author):

The manuscript "Mitochondrial H₂O₂ release does not directly cause genomic DNA damage" by van Soest et al. from the Dansen laboratory investigates whether H₂O₂ produced at physiological-relevant levels at the outer mitochondrial membrane is able to cause DNA damage and results in consequences for cell fate. Van Soest and colleagues employ localized H₂O₂ by the enzyme DAAO, directly at the chromatin and at the outer mitochondrial membrane, to probe consequences of ROS for cell fate (proliferation/senescence/cell death) and DNA damage. In agreement, with previous studies (e.g., PMID: 35146782, PMID: 32130885) which used targeted DAAO inside mitochondria, they find that also H₂O₂ produced at the outer mitochondrial membrane cannot be detected by a nuclear localized H₂O₂ sensor Hyper7 (<10 mM d-ala). Consequently, only H₂O₂ produced at the chromatin induced DNA damage induced a largely p53 dependent 4N cell cycle arrest, whereas mitochondrial membrane ROS resulted at high concentration in cell death, that in part, could be rescued by inhibiting ferroptosis. Therefore, the authors conclude that "Mitochondrial H₂O₂ release does not directly cause genomic DNA damage".

Aberrations in the production of reactive oxygen species (ROS) often due to metabolic changes are common in cancer cells and are thought to contribute to genomic instability and tumorigenesis. Thus, addressing possible pathological consequences of mitochondrial-derived ROS as presented in this manuscript is an important subject relevant to multiple research fields in life sciences. The current version, however, has several issues that should be addressed:

Major:

1) ROS production in the mitochondria occurs in the matrix and inner membrane space (IMS), and subsequently must pass one or two mitochondrial membranes to reach the cytoplasm. Here, the authors produce H₂O₂ at the outer mitochondrial membrane by using TOM20. Can the authors explain why a TOM20 localized probe is better suited to investigate the consequences of mitochondrial produced ROS for cell fate than DAAO localized inside the mitochondria (IMS or matrix? – i.e., PMID: 35146782 show that nuclear Hyper7 detects ROS produced inside the mitochondria by 8 mM d-ala.

2) The authors promote a new approach by using Seahorse measurements to determine and H₂O₂ production by DAAO and relate it to the levels of endogenously produced H₂O₂. The study describing and verifying this approach “will be posted within a few weeks on BioRxiv”. Without this data out the authors must provide evidence that using OCR is indeed a valid approach to compare the activities of different DAAO probes and cell lines in the presented manuscript. E.g., even though the H2B and TOM20 reporter cell line consume oxygen at the same rate they oxidize Hyper7 to a very different extent (Figure 1a/c), which either can reflect differences in reduction potential of nucleus and cytoplasm or different expression/activity levels of DAAO. To this reviewer’s opinion the Prdx2 data presented in Figure S1a suggest that the ratio of Prdx dimer to monomer is significantly higher in the TOM20 cell line than in the H2B line. A problem with the presented data is that the steady-state ratio in L-ala of Prdx dimer to monomer already strongly differs: judged by eye close to 1:1 in TOM20 cells (and here the dimer appears run as two bands?) compared to a much lower ratio in H2B cells. Normalizing to the ratio to L-ala as performed by the authors naturally alleviates this difference and thus might lead to miss-leading interpretations. Doesn’t this indicate that based on Prdx2 the redox state of both reporter lines already differs in the first place (a consequence of clonal differences?) and if yes, how does this affect subsequent interpretations?

Since the authors used clonal cell lines derived from random integration of DAAO as opposed to isogenic cell lines (by integration into a FRT site), they should compare DAAO expression between cell lines by WB and ideally repeat key experiments using more than one clone of each line.

3) All experiments are performed under atmospheric oxygen concentration, which are much higher than what RPE cells experience in physiological conditions. As the occurrence of DNA damage and senescence is strongly linked to the concentration of oxygen the cells are exposed to the authors should repeat key experiments of figure 3 and 4 under a physiological pO₂ for RPE cells to substantiate their claim that “Mitochondrial H₂O₂ release does not directly cause genomic DNA damage”. Ideally, one would use an animal model to confirm this important message, but this is beyond the scope of the current study.

4) The crystal violet experiments reading out viability/proliferation presented in figures 1E and S2 only cover 24h, whereas the bulk of key experiments showing differences in in both lines in figures 3, 4, and 5 were performed after 48h d-ala exposure. Since the response to D-ala is clearly time and concentration dependent (Figure S2) it is important to provide the corresponding crystal violet data also after 48h to put data from figures 3-5 into context.

5) The authors conclude that H₂O₂ released from mitochondria is unable to trigger DNA damage unlike H₂O₂ that is produced in the nucleus. Measuring directly DNA oxidation in a, e.g. by detection 8-oxoguanine or mutations by sequencing approaches should be performed to fully support that

mitochondrial ROS does not produce oxidative DNA damage.

Minor

1) To promote their OCR based approach to measure DAAO activity the authors argue that Hyper7 does not directly measure H₂O₂ (nor that detecting it by OCR..) but rather detects the combined rate of redox. But is this not the relevant parameter that determines molecular cellular consequences of ROS e.g., DNA damage etc.? To my opinion such claims are better suited to the upcoming method paper than this biological study.

2) The introduction should include 8-oxoguanine as a potential key target of ROS linked DNA damage/mutations.

3) It is not discussed whether alternate physiological sources of intracellular ROS (i.e., ER) might produce oxidative DNA damage.

4) Citations need to be included on Page 4, Paragraph 2, "The DNA repair protein RAD51 for instance has been shown to be inhibited upon Cysteine oxidation. Treatment of cells with high dose of exogenous H₂O₂ has been shown to activate both ataxia telangiectasia and RAD3-related protein (ATR) and ataxia telangiectasia mutated protein (ATM) dependent DNA damage response (DDR), indicative of single and double strand DNA breaks respectively."

5) Figure 1b: Please provide colour-blind accessible overlays and B&W staining of one-channel pictures for better visualization.

6) 1c: how many biological repeats?

7) 1d: please define which values from 1c were used to compute 1d (average of all points at a given concentration of D-ala?)

8) 2a-d: The Y axis should be at the same scale for all data to facilitate comparisons. How many cells and biological repeats? How big is the heterogeneity in H₂O₂ production in between cells? If significant this may have consequences for subsequent cell fate analyses. Can the authors standardize their Hyper7 measurements by fully oxidizing and reducing Hyper7, e.g., with H₂O₂ and DTT in both cell lines? Is nuclear Hyper7 at steady-state already more oxidized than cytoplasmic Hyper7?

9) 2e: please provide quantification of data and corresponding statistics. How many biological repeats?

10) 3a,b: please provide molecular markers. The loading based on tubulin in 3b is highly different at earlier (WT) and later (p53ko) d-ala concentrations, respectively. How many biological repeats? A positive control for checkpoint activity in 3b is missing in the general absence of a checkpoint response as a technical control.

11) 3d: showing all 100 data points from both experiments would be more informative than showing the mean of two repeats.

12) 4e: The experiment aims to investigate a potential 2N arrest. However, this is done in the background of a protocol that according to fig. 4d already arrest more than 50% of cells in G2. To better detect a G1 phase phenotype pre-synchronization and release from a G1 phase block (e.g., by using Palbociclib) would be a more sensitive approach.

13) 4d, f: Can the authors comment on why basically repeating the same experiment (H2B WT) gives

different results at lower concentrations of d-ala? Notably, the high amount of 4N cells in the control (L-ala) conditions is peculiar.

14) 5a: please provide quantification, statistics, and scale bar. How many biological repeats?

15) 5c: please provide quantifications of TOM20 DAAO cells in the supplement to support claim in results that there is not effect.

16) 5d: The authors state that at 15 mM d-ala massive cell death occurs in TOM20 cells. Considering this statement can the authors please provide information on the gating strategy of their flow cytometry data, especially regarding sub 2N DNA content. Additional supporting data would help strength that the cells indeed undergo ferroptosis, i.e., rescuing the phenotype with antioxidant or measuring hallmarks of ferroptosis (i.e., lipid peroxidation or protein expression of key regulators).

17) S1a,b. Please provide molecular markers, numbers of repeats, see also major point 2.

18) S2a: please provide quantification of data and corresponding statistics. How many biological repeats? Without additional analyses crystal violet staining alone is not a sufficient determinant of viability. Data should include 48h timepoint (see major point 4).

19) please provide quantification of data and corresponding statistics. Data should include 48h timepoint (see major point 4).

Reviewer #4 (Remarks to the Author):

The issue of whether mitochondrial H₂O₂ production induces nuclear DNA damage and contributes to genomic instability is, certainly, worth of investigation, as clear experimental evidence of a causal relationship is, in fact, lacking. Here, the authors try to address the issue using compartment specific D-amino acid oxidase expression to generate H₂O₂ directly in the nucleus or in the cytosolic side of the mitochondrial membrane. While the experimental design is interesting, the authors made several assumptions to justify that their cellular model generates far more mitochondrial H₂O₂ than it would be generated under normal, physiological conditions. Considering that these are cells growing under atmospheric oxygen concentrations, I think this is a fair assumption. Nonetheless, since this is a core issue to their conclusions, I do think it is necessary to have accurate measurements of H₂O₂ concentrations, generated by "normal" mitochondrial metabolism and under their experimental conditions, not only in terms of arbitrary units of fluorescence, but as H₂O₂ concentration, as most of the conclusions of the study are based on the assumption that the DAAO construct is generating unphysiologically high levels of H₂O₂.

Another central point is the time frame. Indeed, most data in the literature comes from bolus H₂O₂ treatments, which are unlikely to happen in vivo. On the other hand, the nuclear DNA damage accumulation linked to mitochondrial H₂O₂ production could be caused over longer periods of time, due to low levels of diffusing H₂O₂ that escape cytosolic antioxidant defenses and do reach the nucleus. It is not clear that their experimental design can detect these low levels of H₂O₂ or even the low-level DNA damage induced under these conditions

Regarding the model, I do believe that some basic characterizations would benefit the interpretation of the results. In figure 1, the authors use Seahorse respirometry to estimate H₂O₂ production via the compartment specific DAAO expression after inhibiting mitochondrial respiration with oligomycin. What are the actual O₂ consumption rate, in nmol O₂/mg protein or /number of cells/time? This information is important to give the reader a better sense of the H₂O₂ concentration being generated.

Under these conditions, what is the contribution of non-mitochondrial oxygen consumption? Also, can they show that there is no changes in uncoupled respiration during the time frame of the experiment?

And even though the cells were grown under high glucose conditions, can they show that intracellular ATP levels do not drop during the experimental time? I think this is important, considering that one of the endpoints analyzed is DDR, which depends on nucleotide availability. And I think it would be important to show, at least in the supplementary data, that that D-Ala concentrations they use are not toxic to the cells in absence of the DAAO transgene.

It might be due to image quality, but the cell survival results shown in Fig. 2E seem a bit different from the results shown in suppl. Fig. 2, where there seems to be a concentration dependent effect, not seen in Fig. 2E. What is the difference? Also, from sup. Fig 2, there seems to be stronger effect on survival of the DAAO-TOMM20 construct, at least at 20 mM D-Ala, not seen in Fig 2E.

In fig. 2C, 5 and 10 mM D-Ala generated roughly similar levels of H₂O₂ as indicated by the HyPer signal in the DAAO-H2B cells. But when looking at gammaH2A levels induced under these conditions, the levels seem very different, at least when measured by WB (Fig. 3A). How could that be reconciled? This is, again, a central issue to the conclusion of this study, once the issue is how much H₂O₂ should reach the nucleus to cause "sufficient" DNA damage to induce the downstream cellular consequences attributed to mitochondrial H₂O₂ production. A minor point with this figure is that there seems to be a faint, but nonetheless noticeable, band in the p53 WB in the p53KO cells!

Still on figure 3, the representative image used for the comet assay does not seem to represent well the results presented in Fig3D, specially for the 20 mM D-Ala condition.

For the experiments described in Fig 4, the authors treated the cells with D-Ala for 48 hr, followed by 24 hr in absence of the D-amino acid. What was the cellular viability after this long treatment? And how about mitochondrial function? Fig 1D showed that for the DAAO-TOMM20, 10 mM D-ala induced a significant increase in OCR, around 40% of oligomycin-sensitive mitochondrial OCR. Under these conditions, it would be reasonable to suppose that there was some mitochondrial damage accumulation that could impact mitochondrial function. But they see no cell cycle arrest, suggesting that cellular energetics is maintained, even when there is significant mitochondrial H₂O₂ production. Considering the implications of these results, I do believe it is important to show that mitochondrial function is not compromised. The same could be said for the induction of senescence, shown in Fig 5A, as several other groups have shown that mitochondrial H₂O₂ production contributes to inducing senescence.

In fig. 2E they show that 20 mM D-ala kills most of DAAO-H2B cells as well as the DAAO-TOMM20. Is this due to ferroptosis as well, as the suggest being the case for the TOMM20 cells? If not, what is the cytotoxicity mechanism?

Point-by point response to the Reviewer's reports on "Mitochondrial H₂O₂ release does not directly cause genomic DNA damage" by Daan van Soest et al.

General response.

We were pleased to read that all four reviewers seemed in general enthusiastic and intrigued by our findings. The reviewers posed some excellent questions that helped us to strengthen our conclusions and the revised manuscript contains many suggested new and control experiments (as well as new contributing authors). The revised manuscript contains several new figure panels and supplemental figures that further underpin and elaborate our interpretations and conclusions. We have now included Whole Genome Sequencing experiments to investigate the induction of DNA mutations downstream of nucleosome- and outer-mitochondrial membrane-derived H₂O₂ production with single base resolution, from which we conclude that mitochondrial H₂O₂ production also does not lead to low levels of nuclear DNA mutations that we could have missed by other detection methods in the previous version of our manuscript. We have also constructed new monoclonal MCF7 cell lines expressing our inducible system to corroborate our key observation in RPE1-hTERT cells. We think that the revised manuscript addresses the points raised with new experimental data and textual changes, and some minor points that we could not address within the time granted for the revision have been addressed in the discussion of the manuscript.

Reviewer #1

DNA damage is often associated with an overabundance of reactive oxygen species (ROS). Since mitochondria are one of the main sources of ROS they have been frequently implied in the induction of DNA damage through the release of ROS, specifically H₂O₂. The authors of this study tackled this hypothesis by genetically engineering cells and equipping them with a D-amino acid oxidase (DAAO) localized either to the nucleus or the cytosolic side of the mitochondrial outer membrane (TOM20-DAAO). Upon addition of D-alanine, DAAO generates H₂O₂. The authors found that hardly any H₂O₂ reaches the nucleus when produced by TOM20-DAAO although DAAOs at both locations produced equal amounts of H₂O₂. As a consequence, there was also no DNA damage, senescence etc. observable. Upon production of extremely high H₂O₂ concentrations at mitochondria, cells were driven to ferroptosis emphasizing this already known mitochondria-ferroptosis axis.

This is a well-conceived and technically sound study that addresses an important question in the field, namely to which extent does the release of mitochondrial H₂O₂ contribute to cellular damage. The authors demonstrate that hardly any H₂O₂ reaches the nucleus when produced by TOM20-DAAO. Based on this main finding the authors can also convincingly demonstrate that NO H₂O₂ also does not lead to DNA damage and does not pose any danger to the cell (essentially all experiments following Figure 2). However, it is already known from previous studies that only minute amounts of H₂O₂ can diffuse away from mitochondria upon its release. This is due to the strong antioxidative systems of the cytosol. Some of these previous findings have even been made employing exactly the same biosensor as used by the authors (e.g. Hohne et al, EMBO J 2022; Koren et al, bioRxiv 2023; de Cubas et al, Antioxidants 2021). From these previous findings and the present study, it is therefore not completely surprising that mitochondrial H₂O₂ does not lead to nuclear DNA damage in healthy cells as even the high D-Ala concentrations used in this study (10-20 mM) lead to nuclear H₂O₂ levels (produced by TOM20-DAAO) that are well below what lead to any damage with nuclear DAAO.

I have one major concern with this study: I am not sure whether this chemo-genetic approach in "healthy" cells really reflects pathological conditions under which NADPH supply might fail (which ultimately drives antioxidative responses in the cytosol and has been shown to allow H₂O₂ to travel much further!) or mitochondria start localizing more perinuclear. Thus, I am also not sure whether with this study any contribution of mitochondrial H₂O₂ to nuclear DNA damage can be excluded.

We thank the reviewer for their recognition of the importance of the work. We agree that maybe within the Redox Biology field, diffusion of mitochondrial H₂O₂ into the nucleus has been deemed somewhat unlikely under normal circumstances based on for instance the papers mentioned by this reviewer. But this notion is certainly not universally accepted, but there are also recent studies that imply the involvement of ROS/H₂O₂ released by mitochondria in the redox regulation of nuclear proteins and processes (e.g. ^{1,2}). Moreover, when we presented the current manuscript at cancer biology or aging biology meetings, the audience is in general very surprised to see that there is no DNA damage downstream of mitochondrial H₂O₂ release. The dogma of mitochondrial respiration-derived mutation and functional decline is still very strong in the (cell) biology and medical communities (and the general public), and we hope that our study can help to put things in a new perspective.

As the reviewer mentions, we showed that under normal conditions H₂O₂ released by mitochondria as modelled by TOM20-DAAO is scavenged before it reaches the nucleus, and that only if sufficiently (but unphysiologically) high levels are produced it can reach the nucleus and induce DNA damage, but the cells will also die. In our now recently published paper on using Seahorse technology to quantify DAAO activity we indeed show that the extent of HyPer7 oxidation depends on the glucose concentration in the media, and glucose is a main driver of reductive capacity. We therefore agree with reviewer that it is an interesting question to test whether H₂O₂ can diffuse into the nucleus and induce DNA damage without triggering cell death when the cytoplasmic antioxidant system is hampered, leading potentially to mutations that can be passed down the lineage.

To test this, we have now included experiments using the Thioredoxin reductase inhibitor Auranofin to block the peroxiredoxin-thioredoxin system, which is the predominant H₂O₂ scavenging system. Auranofin treatment increases the extent of HyPer7 oxidation in the cytoplasm as expected, indicating that H₂O₂ scavenging is indeed impaired (see **new Suppl. Fig S5**). A small fraction of oxidized HyPer7 can now be detected in the nucleus in response to TOM20-DAAO derived H₂O₂ already at 10 mM D-Ala rather than at 20 mM D-Ala in the absence of Auranofin. However, the extent of TOM20-derived H₂O₂ production at 10 mM D-Ala as a substrate is still much higher than what physiologically likely can be achieved. At 5 mM D-Ala the RPE1-hTERT-DAAO^{TOM20} use the equivalent of ~10% of mitochondrial oxygen consumption for H₂O₂ production (Fig 1C,D), which is unlikely to occur based on estimates of maximal ROS production by purified mitochondria³. One should also consider that TOM20-DAAO produces H₂O₂ at the cytosolic side of the mitochondria. If the same amount of H₂O₂ would be produced inside the mitochondria (of course as O₂[•] which is rapidly dismutated to H₂O₂) only a fraction (if any) would make it over the mitochondrial outer membrane, as suggested by the papers mentioned by the reviewer (Hohne et al, EMBO J 2022; Koren et al, bioRxiv 2023; de Cubas et al, Antioxidants 2021). Furthermore, **new Suppl Fig S15** shows that these levels of H₂O₂ production inside the mitochondrial matrix or in the mitochondrial intermembrane space leads to extensive loss of the mitochondrial membrane potential and hampers respiration, and our previous work presenting the OCR-based quantification of DAAO activity⁴ shows that indeed H₂O₂ production by IMS and MLS targeted DAAO is lethal at lower [D-Ala] as compared to RPE1-hTERT-DAAO^{TOM20}. We conclude that even when the reductive capacity of the cytoplasm is severely hampered, only supraphysiological levels of mitochondrial H₂O₂ released from mitochondria can reach the nucleus and damage the DNA.

The other point, whether H₂O₂ released from mitochondria can reach the nucleus when the mitochondria localize perinuclear is also a very interesting one. In the previous version of our manuscript, we already showed that RPE1-hTERT-DAAO^{TOM20} cells remain cycling in the presence of 48 hrs 10 mM D-Ala as judged by BrdU incorporation. In 48 hrs these cells pass through mitosis. During mitosis cells are more compact and the mitochondria are localized more closely to the chromosomes⁵ and there is no nuclear envelope that could function as a barrier for H₂O₂⁶. But we did not observe a cell cycle arrest or DNA damage induction in the RPE1-hTERT-DAAO^{TOM20} cells, despite H₂O₂ production during mitosis. But DNA damage induction could be obscured in these experiments due to the fact that mitotic cells are a minority in unsynchronised populations. To extend these observations, we trapped cells in mitosis with the spindle poison Nocodazole prior to D-Ala treatment, to test whether H₂O₂ released from mitochondria can damage the DNA during prolonged mitosis. **New Suppl Fig S8** shows that DAAO activation in mitotic cells leads to DNA damage checkpoint activation in case of RPE1-hTERT-DAAO^{H2B}, but not RPE1-hTERT-DAAO^{TOM20} cells. These results further strengthen our conclusion that H₂O₂ released by mitochondria does not lead to nuclear DNA damage, even when mitochondria are localized closer to the DNA and the nuclear envelope is absent.

We thank the reviewer for raising these points, because we think the experiments performed in order to address them has strengthened our observations and conclusions. We like to mention that we do not exclude that there is never any (patho) physiological scenario possible where H₂O₂ is released by mitochondria and at levels that can reach the nucleus. But we do think that under a range of conditions this is highly unlikely. In the literature however, mitochondrial respiration is often deemed almost equal to ROS production and subsequent DNA damage, and we hope the reviewer agrees that our data shows that this is certainly not the case. We have updated the discussion section to accommodate these views.

Minor points:

- (1) further Markers for Senescence should be used e.g. Senescence-Associated Secretory Phenotype (SASP).

We are aware that there are more potential markers for senescence, and that there is debate about heterogeneity of positivity for senescence markers even within populations of senescent cells^{7,8}. We think however that the main research question of this manuscript is not whether the induced arrest observed in RPE1-hTERT-DAAO^{H2B} cells is indeed (a certain type of) senescence. Furthermore, the senescence phenotype takes several days to develop and if only a small percentage of cells have not undergone senescence and keep cycling, these will heavily dilute the phenotype by taking over the culture dish.

We have now included quantification of the SA-beta-Gal staining (**new Fig 5B**) as well as further analysed the RPE1-hTERT-DAAO^{H2B} cells that accumulate with 4N DNA with the Fluorescent Ubiquitin-based Cell Cycle Indicator (FUCCI, **new Fig 4F, Suppl Fig S13, Suppl Movie S1A, S1B**) these 4N cells in this case are indeed in G1/0). We have updated the text to state more carefully that based on 1) absence of BrdU positivity, 2) accumulation in G_{0/1} with 4N DNA following mitotic bypass, 3) sustained p21 induction, 4) loss of lamin B1, 5) positivity for senescence associated beta-galactosidase, we conclude that H₂O₂ produced in the proximity of the DNA in RPE1-hTERT-DAAO^{H2B} cells induces a cell cycle arrest that is reminiscent of senescence, which was previously shown to be induced by bolus exogenous H₂O₂⁹. None of the markers are induced in RPE1-hTERT-DAAO^{TOM20} upon D-Ala treatment. (**Updated fig 5D and S14**)

We are certainly interested in the type of senescence that is induced by nuclear H₂O₂ and whether SASP develops and what factors are being secreted. We are working on this with our collaborator Peter de Keizer, but we think this is for now beyond the scope of the manuscript.

- (2) lipid peroxidation at mitochondria should be looked as potential driver of ferroptosis.

From the current literature it seems debated to what extent mitochondrial lipid peroxidation always precedes ferroptosis¹⁰. We agree that our model system might be of use in elucidating the mechanisms underlying ferroptotic cell death: especially to study which localized sources of H₂O₂ could trigger ferroptosis. However, although mitochondrial lipid peroxidation can be measured by for instance MitoPerox, it may not be so straightforward to distinguish whether mitochondrial lipid peroxidation is a driver or a bystander event in the induction of the observed oxidative cell death.

We have performed several experiments with MitoPerox to test whether mitochondrial lipid peroxidation precedes cell death (as measured by DAPI exclusion) in our system. MitoPerox indeed becomes oxidized prior to cell death, but we think our data is too preliminary to draw firm conclusions regarding the causal involvement of mitochondrial lipid peroxidation as a driver of ferroptosis, as it could be that lipids at other locations oxidize as well and/or earlier. The retention of MitoPerox in the mitochondria depends on the membrane potential, which dissipates when the cells die, further complicating interpretation. We do think it will be a very interesting question to thoroughly address in a follow-up study.

In the context of the current manuscript the key observation is that at excessive (and suprphysiological) levels of H₂O₂ production at the outer mitochondrial membrane it can enter the nucleus and cause DNA damage, but that under these conditions the cells die and hence mutations derived from oxidative damage are not propagated. We feel that whether cell death is due solely to ferroptosis and whether it stems (only) from mitochondrial lipid peroxidation is of secondary importance for the main message of the manuscript. Note that cell death can only be rescued in part by ferroptosis inhibitors in our experiments. We have therefore for now rephrased the manuscript to indicate that suprphysiological H₂O₂ produced at the outer mitochondrial membrane induces 'oxidative cell death'.

We have updated the discussion section to include a paragraph on whether the observed oxidative cell death is (only) ferroptosis and the potential involvement of mitochondrial lipid peroxidation.

Reviewer #2

It is well known that ROS can induce DNA damage, stimulate the DDR and contributes to cellular stress and mutagenesis. Given that mitochondrial respiration is a major contributor to cellular ROS, it was assumed that mitochondrial derived ROS is the central contributor to nuclear DNA damage. However, there was a lack of direct evidence supporting this claim. The authors have set up a clever system to directly test this model. They sought to determine whether mitochondrial-derived ROS influences nuclear DNA and cellular responses to DNA damage. Ectopic expression of DAAO, either at the mitochondrial outer membrane (TOM20) or nuclear histones (H2B), allows for inducible ROS production directly in the nucleus or at mitochondria. The authors then used a combination of imaging, flow cytometry and biochemical assays to determine cellular responses to nuclear and mitochondrial derived ROS. The authors central finding is H₂O₂ generated at mitochondria does not affect dsDNA breaks, cell cycle arrest, DNA damage (comet tails), nor cell death (at lower concentrations). These findings are quite surprising. We think that this is an important study and because the authors are going against dogma, we feel that additional experiments are required to shore up the central premise. Overall, we are excited about this work, and feel that these additional experiments would provide better support to their conclusion that mitochondrially derived H₂O₂ does not cause genomic DNA damage.

We thank the reviewer for their excitement about our work and their constructive criticism.

Major comments:

-A major concern is the inconsistency of time points for the assay, and correlating the findings at one time point with findings at another time point. For example, the timing of H₂O₂ production was performed at 9 hours, Prdx2 at 2 hours, cell death assays were performed at 24h, DNA damage at 48 hr. It would be ideal if the authors examined H₂O₂ production closer to the time of the cell behavior? Does the H₂O₂ generation stay high? Does it go down, then how does that relate to cell cycle/DNA damage/etc?

We agree with the reviewer that the used timepoints could be more synchronized. However, please note that timepoints were mostly chosen based on the event or phenotype studied. Cell cycle arrest for instance depends on transcription factor activation, mRNA and protein synthesis and takes more time to develop than immediate effects of H₂O₂ production like Prdx2 and HyPer7 oxidation and kinase-dependent activation of the DNA damage response, and hence a certain level of H₂O₂ production and the observed cellular behaviour at that same time point are not necessarily coupled. Note that the original figures for HyPer7 oxidation were not at 9 hrs but followed for 9 hrs after D-Ala addition. We have now updated these figures to include prolonged (16 hrs and 48 hrs) HyPer7 measurements (**updated Fig 2A-D** and **new suppl Fig S2A**). HyPer7 oxidation stays elevated for 48 hrs but does go down from 10 hrs or so. It is not likely that the substrate runs out and the lower levels of HyPer7 oxidation might be due to cellular adaptation. We have included also more timepoints for other key experiments and readouts (**New suppl Figs S6A, S6B and S9D**) as per the reviewer's suggestion.

-Due to these surprising (but very interesting) results, it is be important to show universality, and we suggest performing key experiments (H₂O₂ production at mitos or nucleus, and examining DNA damage) in at least one other non-transformed cell line.

This is an excellent suggestion. We assumed that with a non-transformed cell line the reviewer means an immortalized but not oncogene-transformed, rather than a primary cell line. Repeating the experiments with primary cells would be very hard to achieve since these cells usually will enter growth arrest upon lentiviral transduction, preventing monoclonal derivation which is necessary to have uniform DAAO activity.

The RPE1-hTert cells used in the manuscript is an immortalized, near-diploid and not oncogenically transformed human cell line (ATCC CRL-4000). Attempts to lentivirally transduce other immortalized (near) wildtype, not oncogenically transformed cell lines to derive monoclonal lines was unfortunately unsuccessful in the time allowed for the revision, probably in part due to incompatibility of the resistance

markers on our vectors. To not further delay the resubmission, we resorted to the human breast cancer line MCF7. We reasoned that it could actually be informative to see whether our observations in RPE1 cells hold true in a cancer cell line. We derived monoclonal MCF7DAAO^{H2B} and MCF7DAAO^{TOM20} and picked clones with equal DAAO activities as determined by the oxygen consumption assay. The MCF7 monoclonals were more sensitive to H₂O₂ induced cell death and these lines died at ~10 mM of D-Ala. But the level of H₂O₂ production in these cells at 10 mM is also far above what is likely physiologically achievable (at 5 mM of D-Ala they already use the equivalent of >10% of oxygen used for mitochondrial respiration for H₂O₂ production). DNA damage (Comet assay and WB for DDR) was induced at 5 mM D-Ala in the MCF7DAAO^{H2B} but not the MCF7DAAO^{TOM20} cell lines, corroborating our RPE1 data. Cell cycle arrest with 4N DNA, p53 stabilization and p21 induction was observed only in the MCF7DAAO^{H2B} cells but were less pronounced than in RPE1 cells, but that might be not unexpected for a cancer cell line. Taken together, mitochondria-derived H₂O₂ also does not directly induce nuclear DNA damage in MCF7 breast cancer cells (**New Suppl Fig S9**).

-Most of the experiments in the supplemental data appear to have only been performed once. Please indicate biological and technical replicates, and statistics. This is also true for Figure 5a-c.

We have provided the missing information. All presented experiments were repeated as indicated.

-Given that this paper is all about the localization of the ROS production, it is important to observe another ROS biosensor to visualize levels (for example, quantification of MitoSOX, or mitochondrially-localized HyPER).

We were admittedly a bit puzzled by this question. MitoSox is a mitochondrially targeted superoxide sensor, that would likely not react with DAAO-derived H₂O₂. We used the H₂O₂ specific HyPer7 sensor localized in the nucleus and in the cytosol, to see whether H₂O₂ generated at the outer mitochondrial membrane reaches the nucleus. If we would localize it to the mitochondrial matrix, we might be able to measure whether DAAOTOM20-derived H₂O₂ makes it into the matrix, but we do not see how that would aid to shed light on our research question whether H₂O₂ *released* from mitochondria *directly* causes damage to nuclear DNA, which is what has often been assumed in literature to happen. The non-reducing prdx2 WB (Suppl Fig S1) already showed that DAAO^{TOM20} induces (at least) the same extent of prdx2 dimerization as DAAO^{H2B}.

It might be that the reviewer means that we should check whether DAAO^{TOM20} derived H₂O₂ affects mitochondrial function. Long term ROS production inside mitochondria may lead to mitochondrial defects. Indeed, we have included new data showing that production of H₂O₂ by DAAO localized in the matrix or the intermembrane space rapidly depolarizes mitochondria and blocks respiration (**new suppl figure S15**). This does not lead to a DNA damage response (**new suppl figure S7**), but we cannot exclude that over longer periods of time these mitochondrial defects *indirectly* contribute to nuclear DNA damage (added to the discussion section). We did not observe mitochondrial depolarization or loss of respiration in RPE1-hTert-DAAO^{TOM20} cells upon D-Ala administration (**new suppl figure S15**), which to us indicates it is a good way to separate mitochondrial H₂O₂ release from other effects of ROS production inside mitochondria.

-The authors are also encouraged to use another cell viability assay that provides more quantitative data. There are several flow-based assays out there, or even direct cell counts.

The previous version of the manuscript already showed PI-exclusion measurements by Flow cytometry to measure viability for some experiments (e.g. figure 5E). We have now included more quantitative data for viability using Propidium Iodide exclusion by Flow Cytometry (e.g. Figs S4B, S9C) and spectrophotometric quantification of the colony assay after redissolving the Crystal Violet stain in acetic acid. (Fig S4A)

Minor Comments:

-In text typo: 'Reviewed in PMID 20064600'

Thank you, this has been fixed.

-Treatment of cells with a high dose of exogenous H_2O_2 has been shown to activate both ataxia telangiectasia and RAD3-related protein (ATR) and ataxia telangiectasia mutated protein (ATM) dependent DNA damage response (DDR), indicative of single and double strand DNA breaks respectively.' – reference needed.

We have included references for H_2O_2 dependent ATM and ATR activation.

-In text typo: 'Therefore, we monitored whether the DDR was activated in RPE1-hTert-DAAOTOM20 and RPE1-HTERT-DAAOH2B lines upon D-Ala treatment in.'

This has been fixed

-A better explanation for how the D-ala-induced ROS production relates to normal levels of ROS is needed. Its hard to interpret what these data mean without understanding this key point, and the text did not make it clear.

We have updated this section to make it clearer.

Reviewer #3

The manuscript "Mitochondrial H_2O_2 release does not directly cause genomic DNA damage" by van Soest et al. from the Dansen laboratory investigates whether H_2O_2 produced at physiological-relevant levels at the outer mitochondrial membrane is able to cause DNA damage and results in consequences for cell fate. Van Soest and colleagues employ localized H_2O_2 by the enzyme DAAO, directly at the chromatin and at the outer mitochondrial membrane, to probe consequences of ROS for cell fate (proliferation/senescence/cell death) and DNA damage. In agreement, with previous studies (e.g., PMID: 35146782, PMID: 32130885) which used targeted DAAO inside mitochondria, they find that also H_2O_2 produced at the outer mitochondrial membrane cannot be detected by a nuclear localized H_2O_2 sensor Hyper7 (<10 mM d-ala). Consequently, only H_2O_2 produced at the chromatin induced DNA damage induced a largely p53 dependent 4N cell cycle arrest, whereas mitochondrial membrane ROS resulted at high concentration in cell death, that in part, could be rescued by inhibiting ferroptosis. Therefore, the authors conclude that "Mitochondrial H_2O_2 release does not directly cause genomic DNA damage".

Aberrations in the production of reactive oxygen species (ROS) often due to metabolic changes are common in cancer cells and are thought to contribute to genomic instability and tumorigenesis. Thus, addressing possible pathological consequences of mitochondrial-derived ROS as presented in this manuscript is an important subject relevant to multiple research fields in life sciences. The current version, however, has several issues that should be addressed:

We thank the reviewer for their enthusiasm, constructive points, and excellent suggestions for improvement.

Major:

1) ROS production in the mitochondria occurs in the matrix and inner membrane space (IMS), and subsequently must pass one or two mitochondrial membranes to reach the cytoplasm. Here, the authors produce H_2O_2 at the outer mitochondrial membrane by using TOM20. Can the authors explain why a TOM20 localized probe is better suited to investigate the consequences of mitochondrial produced ROS for cell fate than DAAO localized inside the mitochondria (IMS or matrix? – i.e., PMID: 35146782 show that nuclear Hyper7 detects ROS produced inside the mitochondria by 8 mM d-ala.

The reviewer is of course right that mitochondrially derived ROS occurs in the matrix and in the intermembrane space. What we investigate in this manuscript is whether the idea that ROS that is released by mitochondria (in the form of H_2O_2), could possibly directly oxidize nuclear DNA and thus contribute to mutations, as is often suggested, or inferred in the (cancer biology and aging) literature. As the reviewer notes, it has been recently suggested that very little, if any H_2O_2 comes out of the mitochondria. We

reckoned that by using DAAOTOM20 we could mimic and titrate H₂O₂ release. If we would have used mitochondrially targeted DAAO, we would have had the problem that it is difficult to assess whether and quantify how much of the H₂O₂ is released to the cytosol. Furthermore, our new data show that H₂O₂ production inside mitochondria impairs mitochondrial function (**New Suppl Fig S15**) making it difficult to study the direct effect of H₂O₂ release in isolation. We have included new data that show that the DDR is also not triggered by matrix (DAAO^{MLS}) and intermembrane space localized DAAO (DAAO^{IMS}).

The fact that Hoehne et al (EMBO J 2022) observe NLS-HyPer7 oxidation at 8 mM D-Ala might be simply due to differences in DAAO expression/activity or cell line specific effects. It is not clear how much H₂O₂ is produced in their system with 8 mM D-Ala and how that compares to our model system or to (patho)physiologically achievable amounts of H₂O₂ release and whether the cells in their experiment survive long term. Our new data in MCF7 cells (**new suppl Fig S9**) also shows that these cells die already at lower [D-Ala], indicating that there might indeed be cell type specific differences.

2) The authors promote a new approach by using Seahorse measurements to determine and H₂O₂ production by DAAO and relate it to the levels of endogenously produced H₂O₂. The study describing and verifying this approach “will be posted within a few weeks on BioRxiv”. Without this data out the authors must provide evidence that using OCR is indeed a valid approach to compare the activities of different DAAO probes and cell lines in the presented manuscript.

That methods paper has now been published in Free Radical Biology and Medicine⁴.

E.g., even though the H2B and TOM20 reporter cell line consume oxygen at the same rate they oxidize Hyper7 to a very different extent (Figure 1a/c), which either can reflect differences in reduction potential of nucleus and cytoplasm or different expression/activity levels of DAAO.

This is exactly one of the reasons why we think measuring DAAO activity is more informative than measuring HyPer7 to estimate how much H₂O₂ is being produced.

To this reviewer’s opinion the Prdx2 data presented in Figure S1a suggest that the ratio of Prdx dimer to monomer is significantly higher in the TOM20 cell line than in the H2B line. A problem with the presented data is that the steady-state ratio in L-ala of Prdx dimer to monomer already strongly differs: judged by eye close to 1:1 in TOM20 cells (and here the dimer appears run as two bands?) compared to a much lower ratio in H2B cells. Normalizing to the ratio to L-ala as performed by the authors naturally alleviates this difference and thus might lead to miss-leading interpretations. Doesn’t this indicate that based on Prdx2 the redox state of both reporter lines already differs in the first place (a consequence of clonal differences?) and if yes, how does this affect subsequent interpretations?

The main purposes of this experiments are 1) to show independently of HyPer7 that the cell lines do produce H₂O₂ 2) to show that the observed difference in DNA damage induction is not due to a higher reductive capacity in the DAAO-TOM20 line. 3) to see whether localized H₂O₂ production overwhelms the Prdx-Trx reductive system and leads to overoxidation of Prdx. We agree with the reviewer that the DAAO-TOM20 line seems to have slightly higher basal levels of Prdx2 oxidation. To us this was reassuring, because it means that even with a lower basal reductive capacity the DAAO-TOM20 line does not show DNA damage at comparable levels of H₂O₂ production. As mentioned earlier, we have used titrations of D-Ala throughout the manuscript and the cell lines with differentially targeted DAAO do not start to phenocopy each other at higher (but sublethal) [D-Ala], suggesting that differences in basal H₂O₂ clearance capacity are not at the basis of the observed differences in the ability to damage DNA when comparing DAAO-TOM20 and DAAO-H2B cell lines. Overoxidation was only observed in response to exogenously added H₂O₂ and not in DAAO-TOM20 or DAAO-H2B cells when treated with the indicated amounts of D-Ala. This suggests that the Prdx-scavenging system is not overwhelmed, but also that DNA damage in the DAAO-H2B line is induced before Prdx becomes (detectably) overoxidized.

Since the authors used clonal cell lines derived from random integration of DAAO as opposed to isogenic cell lines (by integration into a FRT site), they should compare DAAO expression between cell lines by WB and ideally repeat key experiments using more than one clone of each line.

We agree that it is important to compare cell lines with equal DAAO activity at different localizations, but we think that our OCR dependent method is more reliable for this purpose, because it measures enzyme activity. Expression levels are not so informative for comparing activities of differentially localized DAAO enzyme, because the activity also depends on FAD availability as well as uptake and diffusion of oxygen and D-Ala across compartments. We have selected monoclonals that express DAAO at different locations but with comparable activities based on the OCR. Furthermore, if differences in activity were at the basis of the different phenotypes observed in the RPE1-hTert-DAAO^{H2B} and RPE1-hTert-DAAO^{Tom20} lines, we would expect that at higher D-Ala one line would start to phenocopy the other. But at higher D-Ala the RPE1-hTert-DAAO^{Tom20} lines just die, they do not arrest or become senescent. We have now also corroborated our key observations in a different cell line (MCF7, see **new Figure S9**)

3) All experiments are performed under atmospheric oxygen concentration, which are much higher than what RPE cells experience in physiological conditions. As the occurrence of DNA damage and senescence is strongly linked to the concentration of oxygen the cells are exposed to the authors should repeat key experiments of figure 3 and 4 under a physiological pO₂ for RPE cells to substantiate their claim that “Mitochondrial H₂O₂ release does not directly cause genomic DNA damage”. Ideally, one would use an animal model to confirm this important message, but this is beyond the scope of the current study.

Although we do agree that the oxygen tension is important parameter in DNA damage and senescence, there are two practical reasons why we have not tried these experiments: first, our lab is simply not equipped to culture, treat, and passage cells under normoxia/low oxygen. Second, DAAO depends on oxygen to produce H₂O₂. We therefore cannot assume that the differentially localized DAAO also produces equal amounts of H₂O₂ at the same [D-Ala], and we do not see how to assess this (it would require to put the Seahorse machine in low oxygen).

Oxygen tension has been linked to the induction of DNA damage and mutations and senescence: culturing cells under normoxic rather than atmospheric, hyperoxic conditions lowers the C>A mutation rate (likely due to 8-oxo-dG) and delays senescence^{11,12}. Since we have performed our experiments at atmospheric oxygen tension, we therefore think that it is even less likely that mitochondrial H₂O₂ release can contribute to nuclear DNA damage under normoxic conditions. We therefore think the suggested experiments at normoxia would not alter our conclusion. We have added a few sentences on the role of oxygen tension in DNA mutation to the discussion section.

4) The crystal violet experiments reading out viability/proliferation presented in figures 1E and S2 only cover 24h, whereas the bulk of key experiments showing differences in in both lines in figures 3, 4, and 5 were performed after 48h d-ala exposure. Since the response to D-ala is clearly time and concentration dependent (Figure S2) it is important to provide the corresponding crystal violet data also after 48h to put data from figures 3-5 into context.

We have included extra timepoints for several assays including the Crystal Violet based colony assay at 48 hrs, also in response to reviewer 2. See **new suppl Fig S4A**.

5) The authors conclude that H₂O₂ released from mitochondria is unable to trigger DNA damage unlike H₂O₂ that is produced in the nucleus. Measuring directly DNA oxidation in a, e.g. by detection 8-oxoguanine or mutations by sequencing approaches should be performed to fully support that mitochondrial ROS does not produce oxidative DNA damage.

We are very grateful for this excellent suggestion because we think it has strengthened our conclusions substantially. Indeed, the previously presented data using comet assays and the DDR indicate mostly whether single and double strand breaks occur and oxidative modifications like 8-oxo-dG may occur in the absence of breaks in response to (mitochondrially-derived) H₂O₂. Furthermore, DNA mutations may be the result of impaired repair, which could be an indirect effect of H₂O₂, for instance due to oxidative inhibition of repair enzymes. Because quantitative 8-oxo-dG detection has several challenges¹³, and other mutations than C>A (which occurs when 8-oxo-dG is not repaired) may be induced as well, we collaborated with the group of whole genome sequencing and mutational analysis expert Ruben van Boxtel (now co-author). Because the location of the mutations is random, monoclonals have to be derived and expanded for DNA isolation and sequencing following mutation induction. We subjected the cells to 4

rounds of 2 hrs D or L-Ala treatment followed by 22 hrs recovery (**New Fig 4E**). We chose this approach and used the p53 KO background in order to maximize mutation load without killing or arresting the cells. **New Fig 4F** shows that even at these high [D-Ala], no DNA mutations are induced in the DAAO^{TOM20} line, whereas the DAAO^{H2B} line does show many mutations. We therefore (again) conclude that mitochondrial H₂O₂ does not directly induce DNA damage and mutations.

Minor

1) To promote their OCR based approach to measure DAAO activity the authors argue that Hyper7 does not directly measure H₂O₂ (nor that detecting it by OCR..) but rather detects the combined rate of redox. But is this not the relevant parameter that determines molecular cellular consequences of ROS e.g., DNA damage etc.? To my opinion such claims are better suited to the upcoming method paper than this biological study.

We agree that for many cellular redox-dependent processes the redox state is probably more important than the absolute level of H₂O₂. But for oxidative DNA damage (which is primarily induced by hydroxyl radicals derived from H₂O₂), one could envision that it could be induced by a more oxidizing redox state that is due to more oxidants, not due to less reductants (oxidized peroxiredoxin or oxidized glutathione are not very reactive to DNA). We thought it was good to point out that a higher HyPer7 signal can also mean a lower reductive capacity, not necessarily more H₂O₂, although the two are of course coupled. We have rephrased this section and referred to the methods paper.

2) The introduction should include 8-oxoguanine as a potential key target of ROS linked DNA damage/mutations.

We have included 8-oxo-dG and subsequent mutations in the introduction as per the reviewer's suggestion.

3) It is not discussed whether alternate physiological sources of intracellular ROS (i.e., ER) might produce oxidative DNA damage.

We have included a few sentences in the discussion on alternative sources of oxidative DNA damage.

4) Citations need to be included on Page 4, Paragraph 2, "The DNA repair protein RAD51 for instance has been shown to be inhibited upon Cysteine oxidation. Treatment of cells with high dose of exogenous H₂O₂ has been shown to activate both ataxia telangiectasia and RAD3-related protein (ATR) and ataxia telangiectasia mutated protein (ATM) dependent DNA damage response (DDR), indicative of single and double strand DNA breaks respectively."

References have been added.

5) Figure 1b: Please provide colour-blind accessible overlays and B&W staining of one-channel pictures for better visualization.

The figure has been updated to display B&W staining of the single channels. We think that a colour-blind accessible overlay is not necessary here because the localizations can be deduced from the single channel pictures (both the first and last author of this manuscript are red-green color blind...)

6) 1c: how many biological repeats?

This information has now been included

7) 1d: please define which values from 1c were used to compute 1d (average of all points at a given concentration of D-ala?)

This information has now been included

8) 2a-d: The Y axis should be at the same scale for all data to facilitate comparisons. How many cells and biological repeats? How big is the heterogeneity in H_2O_2 production in between cells? If significant this may have consequences for subsequent cell fate analyses. Can the authors standardize their Hyper7 measurements by fully oxidizing and reducing Hyper7, e.g., with H_2O_2 and DTT in both cell lines? Is nuclear Hyper7 at steady-state already more oxidized than cytoplasmic Hyper7?

Thank you for these great suggestions. We have indicated the number of repeats. The curves display the average HyPer7 signal in all cells in view (hundreds of cells).

We have included a new **supplemental figure X** to show the extent of heterogeneity in HyPer7 oxidation in response to DAAO activation (it is not possible to measure heterogeneity in H_2O_2 production between cells as the reviewer proposes). None of the (hundreds of) imaged RPE1-hTert-DAAOTOM20 cells display more nuclear HyPer7 oxidation at 10 mM of D-Ala compared to 10 mM L-Ala.

All HyPER7 measurements have now been standardized by setting the response to 200 micromolar bolus H_2O_2 to 100% oxidation (these measurements were for many experiments already performed but not included in the original manuscript). We have noted that DTT can occasionally increase the HyPer7 signal in our experiments for some reason. But Hoehne et al. showed that the oxidation of HyPer7 under unstressed conditions is very low and hardly affected by further reduction¹⁴ and we have used HyPer7 signal prior to D-Ala treatment as a baseline value.

9) 2e: please provide quantification of data and corresponding statistics. How many biological repeats?

This has now been included in **new figure S4A**.

10) 3a,b: please provide molecular markers. The loading based on tubulin in 3b is highly different at earlier (WT) and later (p53ko) d-ala concentrations, respectively. How many biological repeats? A positive control for checkpoint activity in 3b is missing in the general absence of a checkpoint response as a technical control.

Molecular weight markers have been added. We agree the loading or transfer is somewhat off for tubulin, but since the total levels of Chk1 and Chk2 are equal we think this does not affect our conclusion. All experiments have been performed with (at least) three biological replicates with similar setup. A typical result is shown. The positive control in 3b was there but it had been cropped off and we have now shown it in the revised figure (a very good suggestion indeed).

11) 3d: showing all 100 data points from both experiments would be more informative than showing the mean of two repeats.

We have now included a third replicate and display all data points in one graph along with the mean of the three repeats.

12) 4e: The experiment aims to investigate a potential 2N arrest. However, this is done in the background of a protocol that according to fig. 4d already arrest more than 50% of cells in G2. To better detect a G1 phase phenotype pre-synchronization and release from a G1 phase block (e.g., by using Palbociclib) would be a more sensitive approach.

We apologize if this was not explained clearly. Most cells in culture are generally in G1 (2N DNA). To test whether these cells are arrested in G1 or just happen to be in G1 but still cycling we add Nocodazole during the last 16 hrs of the 48 hrs D-Ala treatment. This will trap (most of the) cycling cells in mitosis (4N DNA) as indicated in **new suppl Fig S8a**. Induction of a G1 block prior to Nocodazole treatment would therefore increase the 2N peak, but we do not observe this and conclude that a 2N G1 block is not induced (for the DAAO^{TOM20} cells we also see in figure 4a that they keep incorporating BrdU and hence are still cycling. We apologise but do not see how the proposed experiment with Palbo release would be more informative than the used approach (which is used more often to detect a G1 arrest¹⁵. In our experience, RPE1 cells do not

easily recover from a Palbo release and part of the cells will remain in G1, which will interfere with the detection of a potential H₂O₂-dependent G1 arrest.

13) 4d, f: Can the authors comment on why basically repeating the same experiment (H2B WT) gives different results at lower concentrations of d-ala? Notably, the high amount of 4N cells in the control (L-ala) conditions is peculiar.

We have moved the original figure 4f to the supplement (**now figure S12**), since figure 4b already shows that the H2B-DAAO cells remain in cycle upon D-Ala treatment in the p53 KO background. There was quite some time (months) between the experiments that resulted in original figure 4d and 4f. RPE1 cells are contact-inhibited and if they are plated too dense, they will arrest which affects cell cycle profile. In the case of the H2B-DAAO cells, this happens more easily in the negative control (L-Ala), because they stay in cycle whereas the D-Ala treated cells arrest. Small differences in plating numbers and different experimenters for the experiments in 4d and original 4f may be at the basis of this difference. It is therefore important to always take along a negative control. But we think that the reviewer will agree that these variations between experiments do not affect our conclusions.

14) 5a: please provide quantification, statistics, and scale bar. How many biological repeats.

We apologize for the omission; this information has been added.

15) 5c: please provide quantifications of TOM20 DAAO cells in the supplement to support claim in results that there is not effect.

The requested experiment has been included in **updated Fig 5D and new Suppl Fig S14**. We thought that since the DAAO-TOM20 cells stay in cycle, remain BrdU positive and do not show SA-beta-Gal it was not necessary to further analyze senescence markers, but we agree it is a nice extra control for the specific induction of these markers in the DAAO-H2B cells.

16) 5d: The authors state that at 15 mM d-ala massive cell death occurs in TOM20 cells. Considering this statement can the authors please provide information on the gating strategy of their flow cytometry data, especially regarding sub 2N DNA content. Additional supporting data would help strength that the cells indeed undergo ferroptosis, i.e., rescuing the phenotype with antioxidant or measuring hallmarks of ferroptosis (i.e., lipid peroxidation or protein expression of key regulators).

All Flow cytometry gating strategies have now been included in Suppl Fig S16. Cell death was assessed by PI-exclusion of live cells, not by sub-2N content in cell cycle profiles. We apologize that we did not make this clear.

As indicated in response to Reviewer 1, we think that although it is an interesting question whether the observed cell death is (only) ferroptosis, this is not the main point of this manuscript. The response to Reviewer 1 has been (partly) copied below:

In the context of the current manuscript the key observation is that at excessive (and supraphysiological) levels of H₂O₂ production at the outer mitochondrial membrane it can enter the nucleus and cause DNA damage, but that under these conditions the cells die and hence mutations derived from oxidative damage are not propagated. We feel that whether cell death is due solely to ferroptosis [...] Note that cell death can only be rescued in part by ferroptosis inhibitors in our experiments. We have therefore for now rephrased the manuscript to indicate that supraphysiological H₂O₂ produced at the outer mitochondrial membrane induces 'oxidative cell death'. We have updated the discussion section to include a paragraph on whether the observed oxidative cell death is (only) ferroptosis and the potential involvement of mitochondrial lipid peroxidation.

17) S1a,b. Please provide molecular markers, numbers of repeats, see also major point 2.

We apologize for the omission; this information has been added.

18) S2a: please provide quantification of data and corresponding statistics. How many biological repeats? Without additional analyses crystal violet staining alone is not a sufficient determinant of viability. Data should include 48h timepoint (see major point 4).

We apologize for the omission; this information has been added. A **new suppl Fig 4A** has been added showing the quantified viability at 48 hrs.

19) please provide quantification of data and corresponding statistics. Data should include 48h timepoint (see major point 4).

We are not sure what figure this comment refers to but we think it has been addressed at minor point 18.

Reviewer #4

The issue of whether mitochondrial H_2O_2 production induces nuclear DNA damage and contributes to genomic instability is, certainly, worth of investigation, as clear experimental evidence of a causal relationship is, in fact, lacking. Here, the authors try to address the issue using compartment specific D-amino acid oxidase expression to generate H_2O_2 directly in the nucleus or in the cytosolic side of the mitochondrial membrane. While the experimental design is interesting, the authors made several assumptions to justify that their cellular model generates far more mitochondrial H_2O_2 than it would be generated under normal, physiological conditions. Considering that these are cells growing under atmospheric oxygen concentrations, I think this is a fair assumption. Nonetheless, since this is a core issue to their conclusions, I do think it is necessary to have accurate measurements of H_2O_2 concentrations, generated by “normal” mitochondrial metabolism and under their experimental conditions, not only in terms of arbitrary units of fluorescence, but as H_2O_2 concentration, as most of the conclusions of the study are based on the assumption that the DAAO construct is generating unphysiologically high levels of H_2O_2 .

We have included this information in the manuscript. The absolute *concentration* of H_2O_2 generated is difficult to assess as this depends also on clearance rates and volume around the localized DAAO considered. But we have provided an estimate of the *amount* of H_2O_2 that is produced per cell (see also below). With respect to the assumption that the DAAO construct is generating unphysiologically high levels of H_2O_2 : we base this also on the relative oxygen consumption by DAAO compared to basal mitochondrial respiration prior to oligomycin addition. Indeed, if basal respiration in these cells was low, it might have been not correct to state that DAAO-dependent oxygen consumption is very high compared to mitochondrial respiration. But our **new Fig S15D** shows that in our RPE1-hTERT-DAAO^{TOM20} cells basal mitochondrial respiration equals about ~75% of maximal respiration. Furthermore, in the RPE1-hTERT-DAAO^{TOM20} cells H_2O_2 is produced at the cytosolic side of the outer mitochondrial membrane, whereas H_2O_2 produced in the ETC would have to cross the mitochondrial membrane(s), which has been shown not to happen to a great extent in a variety of organisms (e.g. ^{14,16-19}). We therefore think our statement that already at 5 mM D-Ala DAAO dependent H_2O_2 production (~0.4 pmol/min/cell) and release into the cytoplasm is already much higher than what mitochondria could normally achieve holds true.

Another central point is the time frame. Indeed, most data in the literature comes from bolus H_2O_2 treatments, which are unlikely to happen in vivo. On the other hand, the nuclear DNA damage accumulation linked to mitochondrial H_2O_2 production could be caused over longer periods of time, due to low levels of diffusing H_2O_2 that escape cytosolic antioxidant defenses and do reach the nucleus. It is not clear that their experimental design can detect these low levels of H_2O_2 or even the low-level DNA damage induced under these conditions.

Since we cannot detect DNA damage or H_2O_2 in the nucleus even at artificially high production rates by DAAO-TOM20 for up to 9 hrs (16 hrs in new figures), we reasoned that low levels of H_2O_2 certainly do not make it into the nucleus. But like noted by Reviewer 3, this reviewer is right that we might miss low levels of DNA damage, especially since we had focussed on detection methods that rely on DNA strand breaks. We have therefore now included whole genome sequencing data, that show that even upon repeated high levels of DAAO^{TOM20} activation no point mutations are induced that would be the result of for instance

unrepaired 8-oxo-dG. Activation of DAAO^{H2B} does result in many single nucleotide alterations. We are grateful for this suggestion because we think these new experiments greatly strengthen our conclusions.

Regarding the model, I do believe that some basic characterizations would benefit the interpretation of the results. In figure 1, the authors use Seahorse respirometry to estimate H₂O₂ production via the compartment specific DAAO expression after inhibiting mitochondrial respiration with oligomycin. What are the actual O₂ consumption rate, in nmol O₂/mg protein or /number of cells/time? This information is important to give the reader a better sense of the H₂O₂ concentration being generated.

We have included this information in the manuscript. At 5 mM D-Ala the oxygen consumption is ~0.4 fmol O₂ cell⁻¹ min⁻¹ above baseline, and hence the H₂O₂ production is ~0.4 fmol H₂O₂ cell⁻¹ min⁻¹.

Under these conditions, what is the contribution of non-mitochondrial oxygen consumption?

The remaining oxygen consumption rate after Oligomycin treatment is ~25%, although we apologize that we do not completely grasp the relevance of this question for the interpretation of the data.

Also, can they show that there is no change in uncoupled respiration during the time frame of the experiment? And even though the cells were grown under high glucose conditions, can they show that intracellular ATP levels do not drop during the experimental time? I think this is important, considering that one of the endpoints analyzed is DDR, which depends on nucleotide availability.

We think that it is not the DNA damage *response* but DNA damage *repair* that depends on nucleotide availability, and the DDR has in fact been shown to be sustained in case repair is hampered due to nucleotide shortage (see e.g. ²⁰). Of course, the DDR kinases depend on ATP, but it is unlikely that [ATP] becomes limiting for kinase activity when the DAAO-TOM20 cells at the same time remain in cycle (Fig 4A). We analysed whether mitochondrial membrane potential (TMRM) or respiration (Seahorse) are affected in the DAAO-TOM20 cells when treated with 10 mM D-Ala for 24 hrs. This was not the case (see **new suppl fig S15**). We have also extended these experiments using DAAO-IMS and DAAO-MLS, which are localized in the intermembrane space and the matrix respectively. H₂O₂ production at these sites does impair mitochondrial function starting at 2.5 mM of D-Ala, but we also do not see evidence of activation of the DDR in these lines (**new suppl fig S7, S15**). The observation that DAAO-TOM20 derived H₂O₂ does not affect mitochondrial function enables to separate the effects of mitochondrial H₂O₂ release (which is often suggested to be a source of DNA damage) from effects of mitochondrial H₂O₂ production on mitochondrial function.

Collectively, we think that we can conclude that mitochondrial H₂O₂ release does not contribute to DNA damage even at supraphysiological levels because:

1) When H₂O₂ is produced at the outer membrane by DAAO-TOM20 it is only detected in the nucleus when an equivalent of ~40% of basal mitochondrial respiration is used by DAAO for H₂O₂ production, whereas literature suggests that a maximum of 2% of mitochondrial oxygen consumption ends up as Superoxide and two superoxide anions yield one molecule of H₂O₂ after rapid dismutation.

2) DAAO-TOM20 is at the outer mitochondrial membrane, mimicking what happens when all mitochondrially produced H₂O₂ is released into the cytoplasm. Normally mitochondrial H₂O₂ is however produced in the matrix and the intermembrane space. Only a fraction of this has been shown to be released and only under specific conditions. Mitochondrial function is completely lost at lower levels of D-Ala than those needed for DAAO-Tom20 derived H₂O₂ to reach the nucleus and lower [D-Ala] are already lethal to DAAO-MLS and DAAO-IMS cells as shown in our now published methods paper on the use of OCR as a measure for DAAO activity ⁴.

3) In response to reviewer 1 we have performed experiments in the presence of Auranofin to deplete the thioredoxin-dependent H₂O₂ scavenging system. Even when the reductive capacity is severely limited DAAO-TOM20 derived H₂O₂ cannot be detected in the nucleus when produced at supraphysiological levels (**new figure S5**)

Note that, as we stated in the discussion section, we do not fully exclude that prolonged enhanced mitochondrial ROS production could lead to nuclear DNA mutation. But we think that it is unlikely that this is because of ROS *released* from the mitochondria traveling to the nucleus. Prolonged mitochondrial ROS

production may impair mitochondrial function (as we show in **new figure S15**), which may in the long run indirectly damage DNA or impair the repair of damaged DNA because of metabolic stress or for instance hampered FeS cluster biogenesis (some DNA repair enzymes are FeS cluster proteins). We point this out in the discussion section of the manuscript.

And I think it would be important to show, at least in the supplementary data, that that D-Ala concentrations they use are not toxic to the cells in absence of the DAAO transgene.

This is a great suggestion and we have included a **new figure S4B**.

It might be due to image quality, but the cell survival results shown in Fig. 2E seem a bit different from the results shown in supl. Fig. 2, where there seems to be a concentration dependent effect, not seen in Fig. 2E. What is the difference? Also, from sup. Fig 2, there seems to be stronger effect on survival of the DAAO-TOMM20 construct, at least at 20 mM D-Ala, not seen in Fig 2E.

We indeed sometimes observe minor differences in the colony assays (which is not unusual with these assays in our experience), which could be due to differences in for instance plating efficiency, cell counts and the experimenter. Sometimes the cells die at 20 mM D-Ala, sometimes at 30 mM D-Ala both of which generate much more H₂O₂ than what likely can be achieved. This is why we include titrations, but we hope that the reviewer agrees that this variability does not affect our main conclusions.

In fig. 2C, 5- and 10-mM D-Ala generated roughly similar levels of H₂O₂ as indicated by the HyPer signal in the DAAO-H2B cells. But when looking at gammaH2A levels induced under these conditions, the levels seem very different, at least when measured by WB (Fig. 3A). How could that be reconciled?

As mentioned, the HyPer7 probe does not measure H₂O₂, but the combined rate of oxidation by H₂O₂ and reduction by the Trx system. There might be a threshold at which H₂O₂ levels produced by DAAO-H2B become sufficiently high to induce DNA damage. We include titrations of D-Ala because the [D-Ala] where this threshold lies might vary slightly between experiments or may be dependent on the used readout. The OCR-dependent measurements show that about twice as much H₂O₂ is produced at 10 mM compared to 5 mM D-Ala. In any case, the message of our manuscript is that H₂O₂ produced at the mitochondria does not induce DNA damage, and we do not see that at any viable concentration of D-Ala in the DAAO-TOM20 line. As mentioned above in response to the other reviewers, if the differences in the observed phenotype would depend only the levels and the localization of H₂O₂ production, one would expect that the DAAO-TOM20 line would start to phenocopy the DAAO-H2B line at higher [D-Ala], which is not the case.

This is, again, a central issue to the conclusion of this study, once the issue is how much H₂O₂ should reach the nucleus to cause “sufficient” DNA damage to induce the downstream cellular consequences attributed to mitochondrial H₂O₂ production.

We hope that the new whole genome sequencing and mutational analysis experiments (Fig 4E, 4F) discussed earlier in response to this reviewer and reviewer 3 have solved the issue of whether our detection method for DNA damage is sensitive enough to pick up low levels of DNA damage.

A minor point with this figure is that there seems to be a faint, but nonetheless noticeable, band in the p53 WB in the p53KO cells!

This is probably a background band. Based on DNA sequencing our CRISPR-Cas9 based targeting was effective, and the DAAO-H2B p53KO cells no longer induce p21 and do not arrest upon high levels of D-Ala. We therefore have no reason to think that there is remaining functional p53.

Still on figure 3, the representative image used for the comet assay does not seem to represent well the results presented in Fig3D, specially for the 20 mM D-Ala condition.

We have included a schematic of how the comet assay has been quantified (**updated fig 3C**). We hope this clarifies this point. An extra replicate has been included as well. All data points of the three replicates are shown, along with the mean of the three biological replicates.

For the experiments described in Fig 4, the authors treated the cells with D-Ala for 48 hr, followed by 24 hr in absence of the D-amino acid. What was the cellular viability after this long treatment?

The viability up until 10 mM D-Ala was almost 100% as can be deduced from **new suppl figure S4A**.

And how about mitochondrial function? Fig 1D showed that for the DAAO-TOMM20, 10 mM D-ala induced a significant increase in OCR, around 40% of oligomycin-sensitive mitochondrial OCR. Under these conditions, it would be reasonable to suppose that there was some mitochondrial damage accumulation that could impact mitochondrial function. But they see no cell cycle arrest, suggesting that cellular energetics is maintained, even when there is significant mitochondrial H₂O₂ production. Considering the implications of these results, I do believe it is important to show that mitochondrial function is not compromised. The same could be said for the induction of senescence, shown in Fig 5A, as several other groups have shown that mitochondrial H₂O₂ production contributes to inducing senescence.

We think the reviewer makes a valid point considering the mitochondrial integrity in context of DAAOTOM20 activation. As already mentioned in our reply to this reviewer at an earlier point, we have now included experiments that assess mitochondrial membrane potential and respiration in response to prolonged H₂O₂ production by DAAO-TOM20. Mitochondrial function is not compromised, unlike when DAAO-MLS or DAAO-IMS is used (**new suppl fig S15**). Please note, that we investigate here whether mitochondrial H₂O₂ release can directly contribute to DNA damage.

In fig. 2E they show that 20 mM D-ala kills most of DAAO-H2B cells as well as the DAAO-TOMM20. Is this due to ferroptosis as well, as the suggest being the case for the TOMM20 cells? If not, what is the cytotoxicity mechanism?

As explained in our response to reviewer 1, we think that it is probably of secondary importance for this study to know what type of cell death is induced in response to, supraphysiological levels of H₂O₂. We think that the more relevant observation is *that* the DAAO-Tom20 cells succumb once so much H₂O₂ is produced that it can be detected in the nucleus, thus preventing outgrowth of cells with potential DNA mutations. We strongly agree it will be interesting to see whether ferroptosis can be initiated by nuclear H₂O₂, and we will investigate this in the future. We also think it is unlikely that the high levels of H₂O₂ in the nucleus at which cells start to die can physiologically be achieved by nuclear localized H₂O₂ producing enzymes like LSD1. That would make studying by what mechanism cells die exactly in these experiments in our view studying something that likely is an artefact. We have updated the discussion section to accommodate these observations.

- 1 Kirova, D. G. *et al.* A ROS-dependent mechanism promotes CDK2 phosphorylation to drive progression through S phase. *Dev Cell* **57**, 1712-1727 e1719 (2022). <https://doi.org/10.1016/j.devcel.2022.06.008>
- 2 Zhang, J. *et al.* Systematic identification of anticancer drug targets reveals a nucleus-to-mitochondria ROS-sensing pathway. *Cell* **186**, 2361-2379 e2325 (2023). <https://doi.org/10.1016/j.cell.2023.04.026>
- 3 Brand, M. D. The sites and topology of mitochondrial superoxide production. *Exp Gerontol* **45**, 466-472 (2010). <https://doi.org/10.1016/j.exger.2010.01.003>
- 4 den Toom, W. T. F. *et al.* Oxygen-consumption based quantification of chemogenetic H(2)O(2) production in live human cells. *Free Radic Biol Med* **206**, 134-142 (2023). <https://doi.org/10.1016/j.freeradbiomed.2023.06.030>
- 5 Martinez-Diez, M., Santamaria, G., Ortega, A. D. & Cuezva, J. M. Biogenesis and dynamics of mitochondria during the cell cycle: significance of 3'UTRs. *PLoS One* **1**, e107 (2006). <https://doi.org/10.1371/journal.pone.0000107>
- 6 Sies, H. Oxidative eustress: On constant alert for redox homeostasis. *Redox Biol* **41**, 101867 (2021). <https://doi.org/10.1016/j.redox.2021.101867>

- 7 Cohn, R. L., Gasek, N. S., Kuchel, G. A. & Xu, M. The heterogeneity of cellular senescence: insights at the single-cell level. *Trends Cell Biol* **33**, 9-17 (2023). <https://doi.org/10.1016/j.tcb.2022.04.011>
- 8 Ashraf, H. M., Fernandez, B. & Spencer, S. L. The intensities of canonical senescence biomarkers integrate the duration of cell-cycle withdrawal. *Nat Commun* **14**, 4527 (2023). <https://doi.org/10.1038/s41467-023-40132-0>
- 9 Chen, Q. M. *et al.* Molecular analysis of H₂O₂-induced senescent-like growth arrest in normal human fibroblasts: p53 and Rb control G1 arrest but not cell replication. *Biochem J* **332** (Pt 1), 43-50 (1998). <https://doi.org/10.1042/bj3320043>
- 10 Zheng, J. & Conrad, M. The Metabolic Underpinnings of Ferroptosis. *Cell Metab* **32**, 920-937 (2020). <https://doi.org/10.1016/j.cmet.2020.10.011>
- 11 Kuijk, E. *et al.* The mutational impact of culturing human pluripotent and adult stem cells. *Nat Commun* **11**, 2493 (2020). <https://doi.org/10.1038/s41467-020-16323-4>
- 12 Chen, Q. M. Replicative senescence and oxidant-induced premature senescence. Beyond the control of cell cycle checkpoints. *Ann N Y Acad Sci* **908**, 111-125 (2000). <https://doi.org/10.1111/j.1749-6632.2000.tb06640.x>
- 13 Collins, A. R., Cadet, J., Moller, L., Poulsen, H. E. & Vina, J. Are we sure we know how to measure 8-oxo-7,8-dihydroguanine in DNA from human cells? *Arch Biochem Biophys* **423**, 57-65 (2004). <https://doi.org/10.1016/j.abb.2003.12.022>
- 14 Hoehne, M. N. *et al.* Spatial and temporal control of mitochondrial H₂ O₂ release in intact human cells. *EMBO J* **41**, e109169 (2022). <https://doi.org/10.15252/emboj.2021109169>
- 15 Medema, R. H., Kops, G. J., Bos, J. L. & Burgering, B. M. AFX-like Forkhead transcription factors mediate cell-cycle regulation by Ras and PKB through p27kip1. *Nature* **404**, 782-787 (2000). <https://doi.org/10.1038/35008115>
- 16 Pak, V. V. *et al.* Ultrasensitive Genetically Encoded Indicator for Hydrogen Peroxide Identifies Roles for the Oxidant in Cell Migration and Mitochondrial Function. *Cell Metab* **31**, 642-653 e646 (2020). <https://doi.org/10.1016/j.cmet.2020.02.003>
- 17 Morgan, B. *et al.* Real-time monitoring of basal H₂O₂ levels with peroxiredoxin-based probes. *Nat Chem Biol* **12**, 437-443 (2016). <https://doi.org/10.1038/nchembio.2067>
- 18 Carmona, M. *et al.* Monitoring cytosolic H₂O₂ fluctuations arising from altered plasma membrane gradients or from mitochondrial activity. *Nat Commun* **10**, 4526 (2019). <https://doi.org/10.1038/s41467-019-12475-0>
- 19 Khan, K. *et al.* Mitochondria-derived reactive oxygen species are the likely primary trigger of mitochondrial retrograde signaling in Arabidopsis. *Curr Biol* (2024). <https://doi.org/10.1016/j.cub.2023.12.005>
- 20 Ludikhuize, M. C. *et al.* Rewiring glucose metabolism improves 5-FU efficacy in p53-deficient/KRAS(G12D) glycolytic colorectal tumors. *Commun Biol* **5**, 1159 (2022). <https://doi.org/10.1038/s42003-022-04055-8>

REVIEWERS' COMMENTS

Reviewer #1 (Remarks to the Author):

the authors have extensively addressed my comments (but also the comments of the other referees) not only in writing but many also experimentally. Collectively, this is a very nice and convincing study. In particular, I would also like to highlight positively that the authors carefully interpret their data!

Reviewer #2 (Remarks to the Author):

The revised manuscript adequately addresses the concerns raised in the initial review. The inclusion of the time courses, the MCF7 data, orthogonal proliferation assays, and statistics strengthen the take home messages of this work. The finding that mitochondrial H₂O₂ release does not seem to induce genomic DNA damage is impactful and sure to be of interest to the metabolism, DNA damage, and cancer communities.

Reviewer #3 (Remarks to the Author):

In their extensive revision, the Dansen laboratory presents a vastly improved manuscript that sheds new light on the role of mitochondrial ROS and its link to cancer and aging - a highly debated and timely issue. They have addressed all key points, either by providing new convincing experimental data, or by reasonable explanations/additional information presented in the text. We congratulate van Soest and colleagues for a well-rounded manuscript that together with the elsewhere published method of OCR-derived H₂O₂ quantifications will open new avenues in redox biology.

Reviewer #4 (Remarks to the Author):

Most of the issues raised have been adequately addressed in the revised version and I think the manuscript is now suitable for publication.

Response to Reviewers related to:

Mitochondrial H₂O₂ release does not directly cause damage to chromosomal DNA.

Daan M.K. van Soest¹, Paulien E. Polderman¹, Wytze T. F. den Toom¹, Janneke P. Keijer¹, Mark J. van Roosmalen², Tim M. F. Leyten¹, Johannes Lehmann¹, Susan Zwakenberg¹, Sasha de Henau¹, Ruben van Boxtel^{2,3}, Boudewijn M. T. Burgering^{1,3}, Tobias B. Dansen¹@

The reviewers had no further points that needed attention. We would like to thank the reviewers for their constructive and helpful comments and enthusiasm.